# A redox-active crosslinker reveals an essential and inhibitable oxidative folding network in the endoplasmic reticulum of malaria parasites

David W. Cobb[1,2¤a], Heather M. Kudyba[1,2¤b], Alejandra Villegas[1,2], Michael R. Hoopmann[3], Rodrigo P. Baptista[2,4], Baylee Bruton[1], Michelle Krakowiak[1,2], Robert L. Moritz[3], Vasant Muralidharan[1,2]*

1 Department of Cellular Biology, University of Georgia, Georgia, United States of America, 2 Center for Tropical and Emerging Global Diseases, University of Georgia, Georgia, United States of America, 3 Institute for Systems Biology, Seattle, Washington, United States of America, 4 Institute of Bioinformatics, University of Georgia, Georgia, United States of America

¤a Current address: Department of Microbiology and Immunology, Vagelos College of Physicians and Surgeons, Columbia University, New York City, New York, United States of America
¤b Current address: Laboratory of Malaria and Vector Research, National Institute of Allergy and Infectious Diseases, National Institutes of Health, Bethesda, Maryland, United States of America
* vasant@uga.edu

**Data Availability Statement:** The mass spectrometry proteomics data have been deposited to the ProteomeXchange Consortium via the

## Abstract

Malaria remains a major global health problem, creating a constant need for research to identify druggable weaknesses in *P. falciparum* biology. As important components of cellular redox biology, members of the Thioredoxin (Trx) superfamily of proteins have received interest as potential drug targets in Apicomplexans. However, the function and essentiality of endoplasmic reticulum (ER)-localized Trx-domain proteins within *P. falciparum* has not been investigated. We generated conditional mutants of the protein *Pf*J2—an ER chaperone and member of the Trx superfamily—and show that it is essential for asexual parasite survival. Using a crosslinker specific for redox-active cysteines, we identified *Pf*J2 substrates as *Pf*PDI8 and *Pf*PDI11, both members of the Trx superfamily as well, which suggests a redox-regulatory role for *Pf*J2. Knockdown of these PDIs in *Pf*J2 conditional mutants show that *Pf*PDI11 may not be essential. However, *Pf*PDI8 is required for asexual growth and our data suggest it may work in a complex with *Pf*J2 and other ER chaperones. Finally, we show that the redox interactions between these Trx-domain proteins in the parasite ER and their substrates are sensitive to small molecule inhibition. Together these data build a model for how Trx-domain proteins in the *P. falciparum* ER work together to assist protein folding and demonstrate the suitability of ER-localized Trx-domain proteins for antimalarial drug development.

PRIDE partner repository with the dataset identifier PXD019100 Project Name: Shotgun MS analysis of human malaria parasite protien PfJ2 and interactors.

**Funding:** The study was funded by NIH/NIAID to V. M. grant number: R01AI130139; to R.L.M. grant numbers: S10OD026936, R01GM087221, and P41GM103533; to D.W.C. and A.V. grant number: T32AI060546 and D.W.C. by ARCS Foundation./ National Institutes of Health (https://www.nih.gov); Advancing Science in America (ARCS) Foundation (https://arcsfoundation.org)/ The funders had no role in study design, data collection and analysis, decision to publish, or preparation of the manuscript.

**Competing interests:** The authors have declared that no competing interests exist.

## Author summary

One of the leading and persistent causes of childhood mortality in the world is malaria, which is caused by parasites from the genus *Plasmodium*. Unfortunately, the parasite has developed resistance to all available drugs, making the discovery of new drug targets and potential small molecule inhibitors of essential parasite biology a top priority. A critical pathway required for many different biological processes in the parasite is oxidative folding which requires members of the Thioredoxin (Trx) superfamily of proteins. But we know almost nothing about the function and essentiality of Trx-domain proteins that localize to the endoplasmic reticulum, the origin of the secretory pathway, within *P. falciparum*. Here we show that a network of Trx-domain containing proteins function together and are essential for parasite survival within human red blood cells. Further, we identify a small molecule inhibitor of the redox activities of these Trx-domain containing proteins. This study demonstrates the suitability of this pathway for future antimalarial drug development.

## Introduction

Today, the majority of the world's population lives at risk for contracting malaria, a disease caused by eukaryotic parasites of the genus *Plasmodium*, with *P. falciparum* causing the most severe forms of the disease [1]. In 2018, the world saw approximately 228 million cases of malaria resulting in more than 400,000 deaths. These numbers reflect a concerted effort to combat malaria in the past few decades, but progress has stagnated, with the numbers of malaria cases/deaths largely unchanged in recent years. A major impediment in the fight against malaria is the rise of drug-resistant—including multidrug-resistant—*P. falciparum* parasites, highlighting the continuous need for research into the biology of this major human pathogen and identification of promising drug targets.

The thioredoxin system and members of the Thioredoxin (Trx) superfamily of proteins have received interest as potential drug targets in Apicomplexan parasites, including in *Plasmodium* [2–5]. Members of the Trx superfamily typically contain at least one Trx domain with a "CXXC" active site. An oxidized Trx domain in which the active site cysteines have formed a disulfide bond can accept electrons to oxidize other proteins, and the sulfhydryl groups of a reduced Trx domain can donate electrons to reduce other proteins. As modulators of protein redox states, members of the Trx superfamily regulate diverse aspects of cellular biology. In *Plasmodium*, members of this superfamily are found in several cellular compartments [6]. Of these, the cytoplasmic Trx system has been most characterized in *Plasmodium* [7]. Within the parasite cytoplasm, Trx Reductase reduces Trx1, which in turn serves as a reductase for other proteins potentially involved in protein synthesis and folding, anti-oxidant stress response, carbohydrate and lipid metabolism, and several other processes [8,9].

An important subset of Trx superfamily members that has received little study in *Plasmodium* are those that localize to the endoplasmic reticulum (ER). Classically, Trx domains in the ER are used to regulate the redox state of cysteines in other proteins, and to facilitate oxidative folding (the formation, reduction, and isomerization of disulfide bonds in newly synthesized proteins) [10]. The *P. falciparum* ER functions in many essential processes during the asexual replication cycle—particularly by serving as the root of the parasite's complex secretory pathway—and ER-localized members of the Trx superfamily likely play critical roles in supporting these functions. *P. falciparum* encodes an ER-localized Hsp40 chaperone with a C-terminal Trx domain known as *Pf*J2, as well as four members of the Protein Disulfide Isomerase (PDI)

family of proteins, which typically use their Trx domains as oxidoreductases to assist folding proteins form their correct disulfide bonds [11,12]. *Pf*J2 bares homology to the mammalian chaperone ERdj5, whose Trx domains serve primarily to reduce other disulfide bonds in the ER, and exogenous overexpression shows ER localization of *Pf*J2 [13–16]. One member of the PDI family, *Pf*PDI8, has been localized to the ER and found able to form and reduce disulfide bonds in folding proteins *in vitro*, including the parasite protein Erythrocyte-Binding Antigen 175 (EBA175) [12,17]. However, these proteins remain largely unstudied in the parasites, and their essentiality for *P. falciparum* survival within the host RBC and functions within the ER have not been investigated.

The ability to reduce disulfide bonds is critical within the oxidative environment of the ER, both for regulating protein function and for allowing correct disulfide pairs to form as proteins are folding [18]. We therefore chose the putative reductase *Pf*J2 for study, and we report here that *Pf*J2 is essential for the *P. falciparum* asexual lifecycle. We identify *Pf*J2 interacting partners and use a chemical biology approach to specifically identify those proteins which may be reduced by *Pf*J2. Surprisingly, these were found to be other Trx-domain proteins: *Pf*PDI8 and *Pf*PDI11. We demonstrate that *Pf*PDI8 is also essential for the asexual lifecycle, and our data suggests that *Pf*J2 and *Pf*PDI8 may work together with the Hsp70 *Pf*BiP to promote protein folding in the ER. Additionally, we demonstrate that the redox interactions between these essential proteins and their substrates are disrupted by a small molecule inhibitor. Together, these data suggest a model in *Plasmodium* for oxidative folding, in which Trx-domain proteins and *Pf*BiP cooperate to ensure proteins reach their native states, and we propose that the oxidative folding process of the *P. falciparum* ER is an exploitable drug target.

## Results

### *Pf*J2 is an essential, ER-resident Hsp40

*Pf*J2 is a putative ER-resident Hsp40 with a C-terminal thioredoxin (Trx) domain (Fig 1A and 1B). It has similarity to a reductase in the mammalian ER which has Trx domains following an Hsp40 J-domain, suggesting PfJ2 may catalyze disulfide bond reduction of client proteins [13,15,16]. To investigate *Pf*J2 function and its potential role in *P. falciparum* oxidative folding, we generated a *Pf*J2 conditional knockdown parasite line using the TetR-*Pf*DOZI aptamer system (referred to as *Pf*J2$^{apt}$ hereafter). In this knockdown system, protein expression is regulated by the presence of anhydrotetracycline (aTc), with knockdown induced by removal of aTc [19] (Fig 1C). Using CRISPR/Cas9 genome editing, we modified the *pfj2* locus to encode a 3xHA-tag immediately upstream of the ER-retention signal, as well as the regulatory aptamer sequences and a cassette to express the TetR-*Pf*DOZI fusion protein (Fig 1D). Correct integration of the construct into the *pfj2* locus was determined by PCR, and expression of HA-tagged *Pf*J2 was demonstrated via western blot (Fig 1D).

Using an indirect immunofluorescence assay (IFA), we assessed the localization of *Pf*J2 throughout the asexual lifecycle (Fig 2A). Co-localization between *Pf*J2 and the ER-marker Plasmepsin V (*Pf*PMV) revealed that *Pf*J2 is in fact an ER-resident protein, consistent with the localization of episomally overexpressed *Pf*J2 previously reported (Fig 2A) [14]. Using highly synchronized parasites, we showed that *Pf*J2 is primarily expressed in the trophozoite and schizont stages, and that during knockdown conditions (removal of aTc), *Pf*J2 expression is reduced (Fig 2B). Importantly, knockdown of *Pf*J2 was found to inhibit expansion of parasites in culture (Fig 2C). Consistent with peak *Pf*J2 expression during the trophozoite/schizont stages, we observed normal development of knockdown parasites in the beginning of the

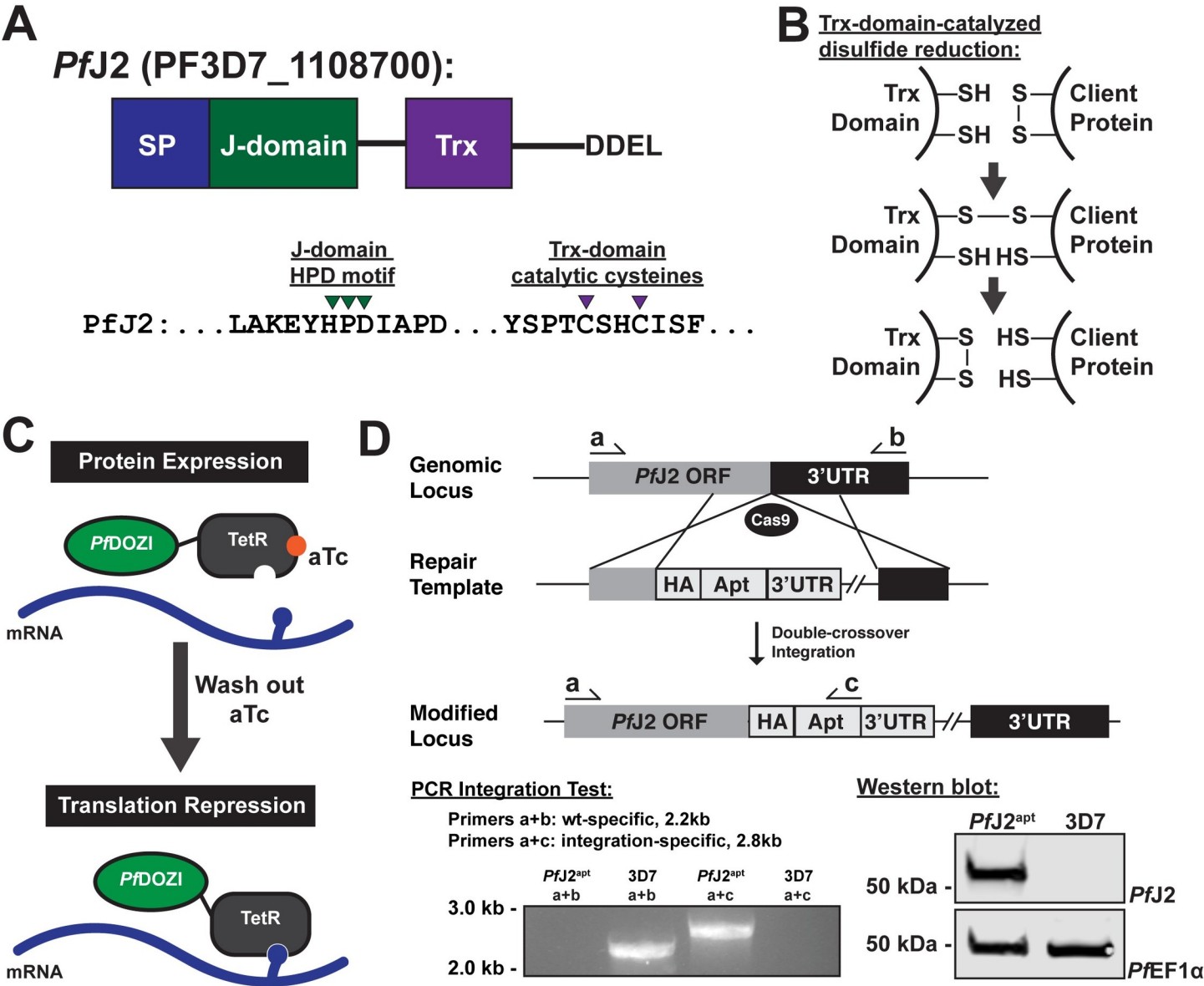

**Fig 1. Generation of *Pf*J2 (PF3D7_1108700) conditional knockdown mutants using CRISPR/Cas9. A)** Predicted domain structure of *Pf*J2 showing signal peptide (SP), Hsp40 J-domain, thioredoxin domain (Trx), and C-terminal ER retention signal. Essential, conserved residues are shown: the J-domain HPD motif is required for Hsp40 activity (i.e., stimulation of Hsp70 ATPase activity), and the Trx-domain CXXC motif is required for redox activity. **B)** Mechanism of disulfide bond reduction catalyzed by Trx-domain active site cysteines. **C)** Regulation of protein expression using the TetR-*Pf*DOZI knockdown system. TetR binds to aptamer sequences present in the mRNA, and *Pf*DOZI localizes the complex to sites of mRNA sequestration, repressing translation. Anhydrotetracycline (aTc) blocks TetR-aptamer interaction. **D)** Schematic of CRISPR/Cas9-mediated introduction of the TetR-*Pf*DOZI knockdown system into the *pfj2* locus. A linearized repair template was transfected, along with a plasmid to express Cas9 and a gRNA, to introduce sequences for a 3xHA tag alongwith *Pf*J2's DDEL ER retention signal, stop codon, and a 3'UTR. Included in the repair template was a cassette to express the TetR-*Pf*DOZI fusion protein and blasticidin deaminase for drug selection. Bottom left: two PCR integration tests were used to amplify either a sequence from only wild-type locus (primers a+b) or the modified locus (a+c). Bottom right: anti-HA western blot. Representative western blot of three biological replicates shown.

asexual life cycle, but the development of these parasites began to slow in the trophozoite stage and they failed to complete schizogony and produce new daughter parasites (Figs 2D and S1). These data demonstrate that *Pf*J2 is a *bona fide* ER-resident protein essential for progression through the *P. falciparum* asexual lifecycle.

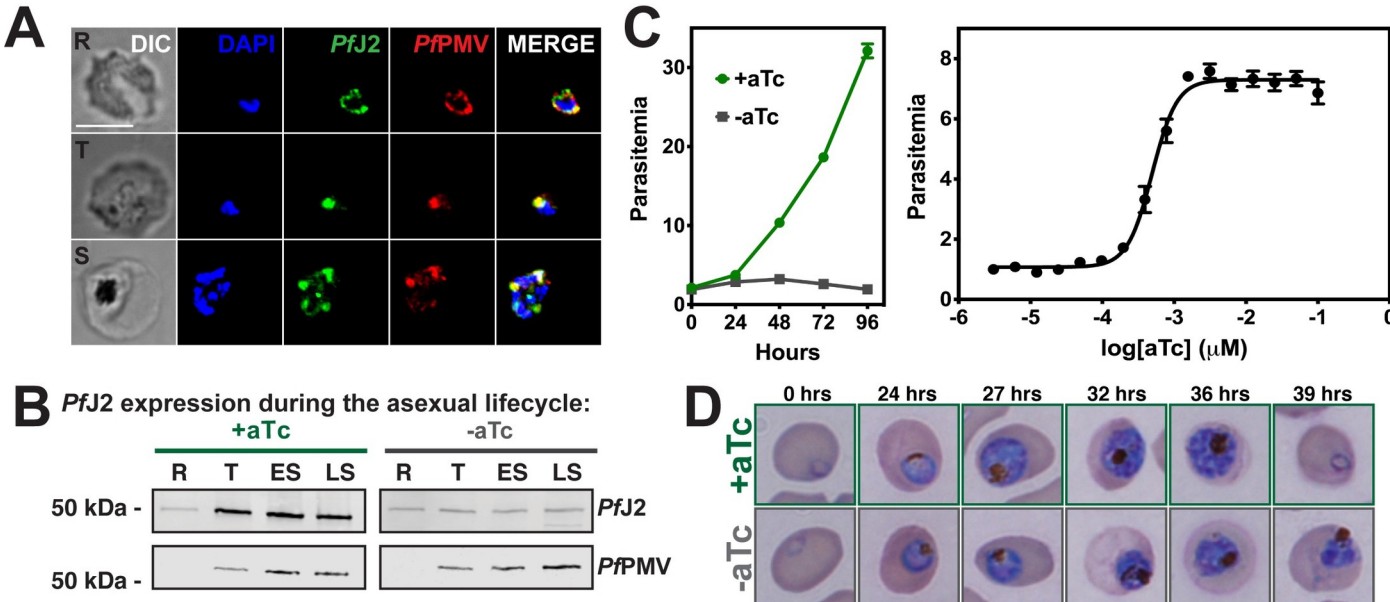

**Fig 2. *PfJ2* is an essential, ER-resident protein. A)** *PfJ2*apt parasites were fixed and stained with DAPI (blue) and with antibodies against HA (green) and the ER-marker *Pf*PMV (red). Ring (R), Trophozoite (T), and Schizont (S) stage parasites are shown. Z-stack Images were deconvoluted and shown as a single, maximum intensity projection. Scale bar represents 5 μm. **B)** Parasites were tightly synchronized to the ring stage (0–3 hours) and split into two conditions: +aTc and–aTc. Samples were taken for western blot analysis at various time points in the life cycle (R = Ring, T = Trophozoite, ES = Early Schizont, LS = Late Schizont). Equal parasite equivalents were loaded into each lane, and membranes were stained with antibodies for HA and *Pf*PMV. Shown is a representative experiment of two biological replicates. **C)** Left: asynchronous parasites were grown in normal (+aTc) or *PfJ2* knockdown (-aTc) conditions, and parasite growth was monitored daily for 96 hours via flow cytometry. Right: asynchronous parasites were grown in a range of aTc concentrations and growth measured at 72 hours via flow cytometry. The aTc $EC_{50}$ was determined to be 0.5 nM. Representative of three biological replicates shown for each growth curve. Each data point represents the mean of three technical replicates; error bars represent standard deviation. **D)** Parasites were tightly synchronized to the ring stage (0–3 hours) and split into two conditions: normal (+aTc) and PfJ2 knockdown (-aTc). Samples from each condition were smeared and field-stained at time points throughout the lifecycle. A representative experiment from three biological replicates is shown.

## *Pf*J2 interacts with essential ER chaperones and proteins in the secretory pathway

As a putative chaperone possibly involved in ER oxidative folding, we reasoned that the essentiality of *PfJ2* is likely related to its ability to interact with other proteins in the ER. We therefore took a co-immunoprecipitation (coIP) approach to identify *PfJ2* interacting partners. *PfJ2* was immunoprecipitated from *PfJ2*apt parasite lysates using anti-HA antibodies, and co-immunoprecipitating proteins were identified by tandem mass spectrometry (MS/MS) analysis. Control parental parasites (lacking HA-tagged *PfJ2*) were also used for immunoprecipitation and analyzed in the same manner. Each co-IP experiment was performed in triplicate, and the abundance of each identified protein was calculated by summing the total MS1 intensities of all matched peptides for each selected protein, and normalizing by the total summed intensity of all matched peptides in the sample (Fig 3A) [20,21]. Because *PfJ2* is an ER-localized protein, we further filtered our list of interacting partners to those containing a signal peptide and/or at least one transmembrane domain (i.e. proteins putatively localized to the ER or trafficked through the secretory pathway). The filter ensures that our list is comprised of proteins that potentially localize or traverse via the cellular compartment where *PfJ2* is located. We identified a stringent list of biologically relevant interacting partners as those proteins which were present in all three *PfJ2*apt coIP experiments, and were at least 5-fold more abundant compared to the controls, as previously described [21] (Fig 3B and 3C). A complete list of all identified proteins is provided in S1 Table.

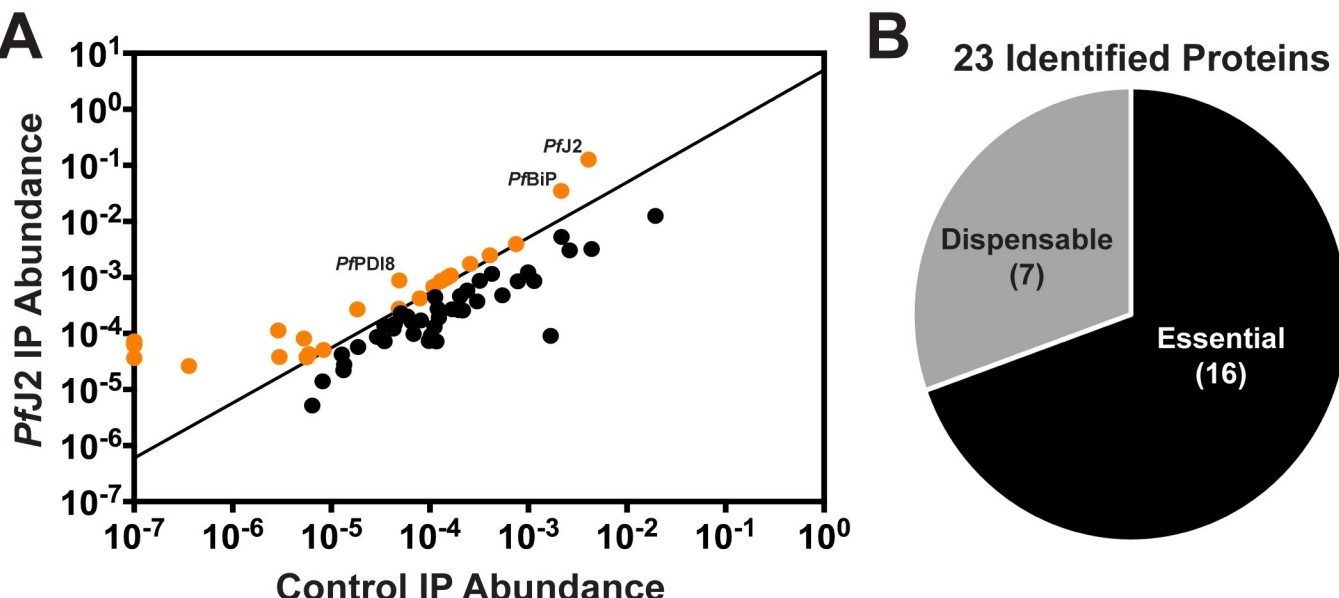

**Fig 3. *Pf*J2 interacts with other essential chaperones, proteins in the secretory pathway. A)** *Pf*J2 was immunoprecipitated from *Pf*J2[apt] parasites using anti-HA antibodies, and co-immunoprecipitated proteins were identified by tandem mass spectrometry analysis. Control, parental parasites were also used for immunoprecipitation and analyzed in the same manner. Each coIP experiment was performed in triplicate, and the abundance of each identified protein was calculated as previously described in Boucher et al. 2018 and Florentin et al. 2020. Biologically relevant candidate proteins of interest were further identified as those potentially in the secretory pathway (predicted to contain a signal peptide and/or transmembrane domains) and those which were present in all three *Pf*J2[apt] replicates and demonstrated a 5-fold enrichment compared to control experiments (shown in orange). **B)** The 23 proteins meeting our strict criteria were assessed against the piggyBac mutagenesis screen performed in Zhang *et al.* 2018, and 16 were predicted to have essential functions in the *P. falciparum* asexual stages. **C)** Identified proteins were categorized by subcellular localization (ER, Rhoptry, Parasite Plasma Membrane [PPM], or Unknown). Also shown are GeneIDs and annotations from PlasmoDB.org, calculated fold-enrichment compared to control experiments, and essentiality as predicted by the piggyBac mutagenesis screen performed by Zhang et al., 2018.

We identified other conserved proteins classically involved in essential ER processes—such as the Hsp70 Binding immunoglobulin Protein (BiP), the Hsp90 Endoplasmin, and the oxido-reductase Protein Disulfide Isomerase (PDI). We further identified proteins that are trafficked through the ER late in the parasite lifecycle and are required for egress and invasion, including *Pf*MSP1 and proteins destined for rhoptries [22–25]. Nearly half of the identified proteins lack empirical evidence for their subcellular localization, and many have no known function. But, given the presence of a signal peptide and/or transmembrane domains, these proteins likely have localizations in the ER, parasite plasma membrane, apicoplast, and other destinations that are part of the secretory pathway. Also of note, approximately two-thirds of the identified proteins are predicted to have essential functions [26]. These data together suggest that *Pf*J2 may work with other ER-resident chaperones to ensure proper folding/functioning of proteins that have essential roles throughout the parasite.

## PfJ2 is a redox-active protein in the *P. falciparum* ER

We next sought to determine whether *Pf*J2 participates in disulfide exchanges with other proteins by chemically trapping these redox partnerships. To this end we employed the bifunctional, electrophilic crosslinker divinyl sulfone (DVSF), which shows remarkable specificity for nucleophilic cysteines, like those present in Trx domain active sites [27]. In *Saccharomyces cerevisiae*, DVSF was used to trap cytosolic Thioredoxin to two proteins known to exchange electrons with the CXXC active site, validating the compound's ability to covalently and irreversibly trap Trx domains to their redox substrates [28]. DVSF was subsequently used to identify substrates of other redox-active proteins containing hyper-reactive cysteines in *S. cerevisiae* and human cells [29,30]. To determine whether DVSF was capable of trapping redox partnerships in *P. falciparum*, we treated *Pf*J2$^{apt}$ parasite cultures with DVSF and isolated proteins for western blot analysis. In the absence of DVSF, *Pf*J2 was detected at approximately 50 kDa, while the addition of DVSF resulted in additional bands containing *Pf*J2 to appear between 100–150 kDa (Fig 4A). To demonstrate specificity of DVSF for nucleophilic cysteines participating in disulfide exchanges, the ER-resident protein *Pf*PMV—which contains 16 cysteines after its signal peptide—was probed for in the same samples and its migration pattern was found to be unaffected by DVSF treatment (Fig 4A). As an additional control, *Pf*J2$^{apt}$ parasite cultures were treated with the sulfhydryl-blocking compound N-ethylmaleimide (NEM) prior to the addition of DVSF. Pre-treatment with NEM resulted in the blockage of cross-linking between *Pf*J2 and its redox partners (Fig 4B). These results indicate that *Pf*J2 likely does participate in disulfide exchange, and that DVSF is a useful chemical tool for trapping redox interactions in the ER of *P. falciparum*.

## *Pf*PDI8 and *Pf*PDI11 are *Pf*J2 redox partners

Having shown the redox activity of *Pf*J2, we next sought to specifically identify those redox partnerships. To identify the proteins trapped to *Pf*J2 by DVSF, cultures were treated with the compound and immunoprecipitation of *Pf*J2 was performed (Fig 4C). As a control, the immunoprecipitation was also performed in parallel using cultures that had not received DVSF treatment. The immunoprecipitated proteins were subjected to separation by SDS-PAGE and visualized using Coomassie (Fig 4C). Two bands between 100–150 kDa, corresponding to those previously detected by western blot, were extracted from the DVSF-treated sample, along with the corresponding areas of the gel in the untreated samples (Fig 4C, perforated boxes). Proteins present in these gel slices were identified via MS/MS analysis. By analyzing the untreated control samples, we were able to remove background proteins and found that the top gel slice primarily contained *Pf*J2, *Pf*PDI8, *Pf*PDI11 and *Pf*BiP, and the bottom slice

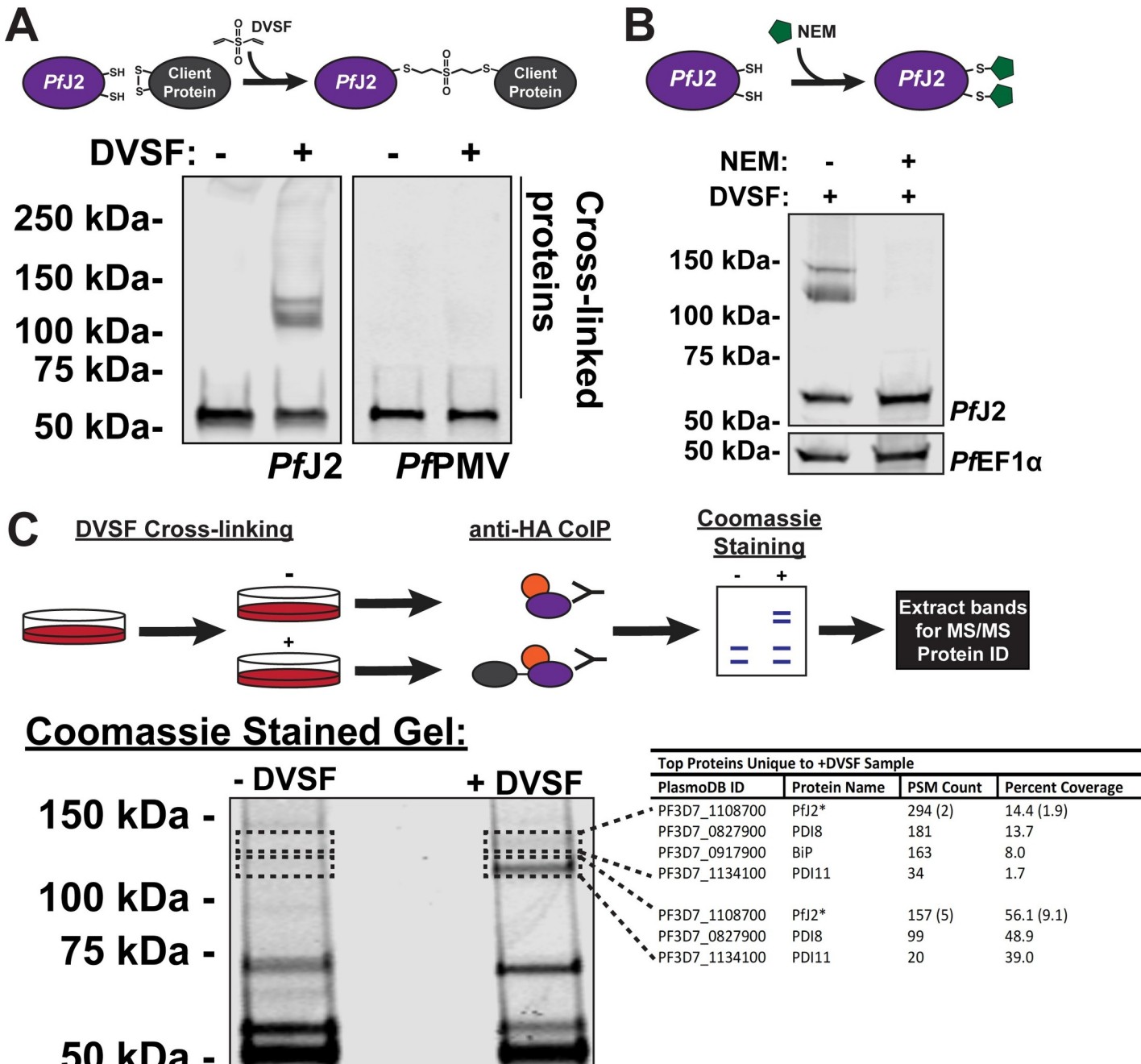

**Fig 4. *Pf*J2 redox partners identified as *Pf*PDI8 and *Pf*PDI11. A)** *Pf*J2[apt] parasites were incubated with 3 mM divinyl sulfone (DVSF) in 1x PBS for 30 minutes at 37˚C, then samples were taken for western blot analysis. Membranes were incubated with antibodies against HA and *Pf*PMV. Representative western blot of three biological replicates is shown. **B)** *Pf*J2[apt] parasite cultures were incubated with 1 mM N-ethylmaleimide (NEM) for 3 hours prior to removal of NEM and addition of 3 mM DVSF as described above. Samples were taken for western blot analysis. Membranes were incubated with antibodies against HA and *Pf*EF1α. Representative western blot of two biological replicates is shown. **C)** *Pf*J2[apt] parasite cultures were evenly split into two conditions: 3 mM DVSF or PBS only for 30 minutes at 37˚C, after which parasite lysates were used for anti-HA immunoprecipitation. Immunoprecipitated proteins were separated via SDS-PAGE and visualized using Coomassie staining. Bands unique to the DVSF-treated sample were extracted, along with the corresponding section of gel in the untreated sample. Proteins were identified by tandem mass spectrometry, and proteins identified in both plus and minus DVSF samples eliminated for further study. The GeneID, protein name, PSM count, and percent coverage for all proteins with more than 1% coverage are shown in the table. A small amount of PfJ2 was identified in the -DVSF samples, and the PSM count and percent coverage is shown in parentheses. One of two biological replicates shown.

contained *Pf*J2, *Pf*PDI8, and *Pf*PDI11 (Fig 4C, table). The complete list of proteins identified in all samples can be found in S2 Table. Because alterations to *Pf*BiP migration during SDS-PAGE after DVSF treatment were not detected (S2 Fig), we chose to focus our attentions on *Pf*PDI8 and -11.

### *Pf*PDI8 and *Pf*PDI11 are redox-active ER proteins

Like *Pf*J2, *Pf*PDI8 and -11 are both predicted members of the Trx superfamily (Fig 5A). PfPDI8 appears to be a canonical PDI, with two Trx domains containing CXXC active sites that likely allow it to carry out disulfide oxidoreductase/isomerase activity. PfPDI11 also has two Trx domains, but each contains an unusual CXXS active site. *Pf*PDI8 has been characterized recombinantly *in vitro*, but the functions of both *Pf*PDI8 and -11 remain unstudied in

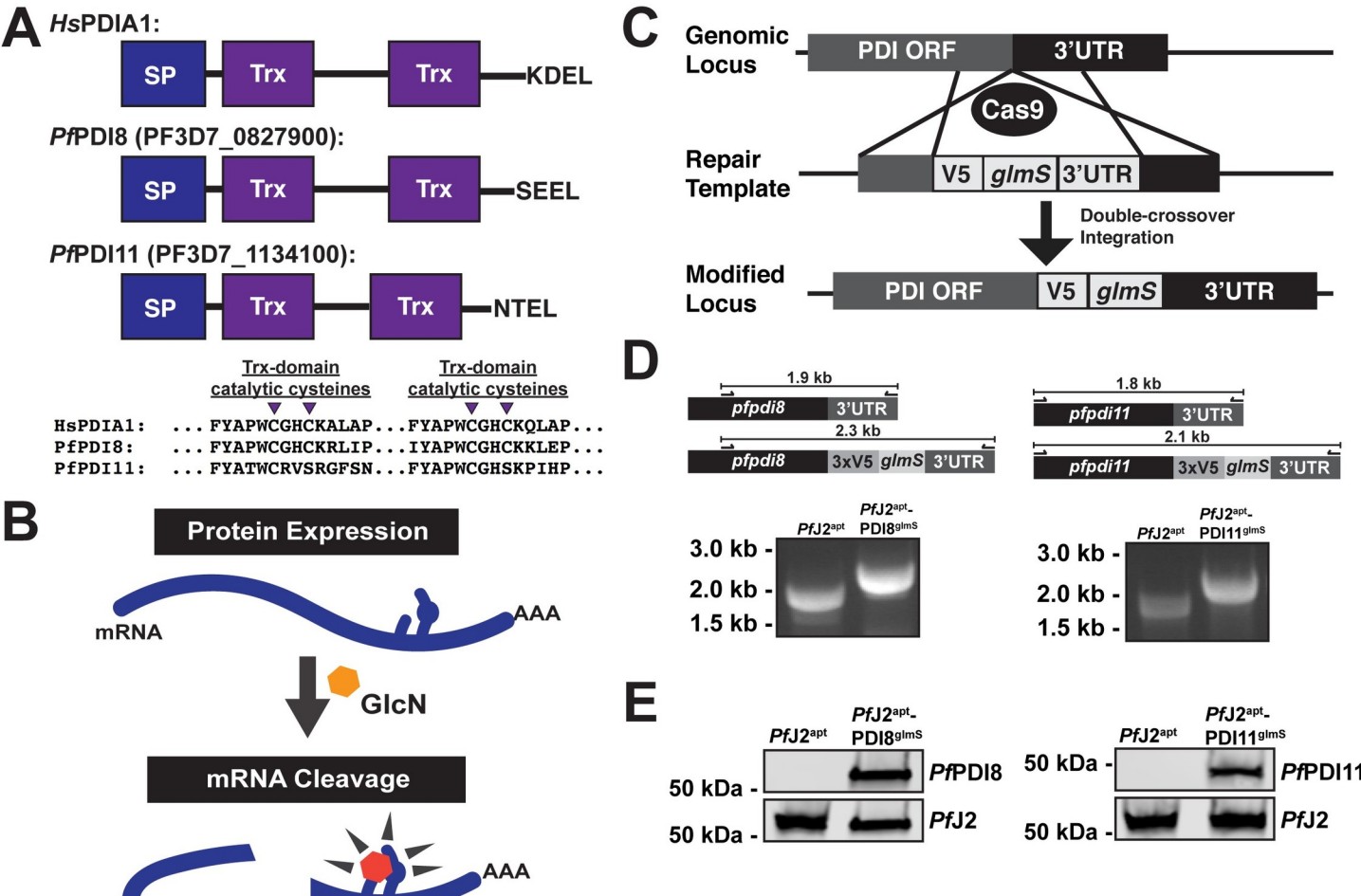

**Fig 5. Generation of *Pf*PDI8 (PF3D7_0827900) and *Pf*PDI11 (PF3D7_1134100) conditional knockdown mutants using CRISPR/Cas9. A)** Predicted domain structure of *Pf*PDI8, *Pf*PDI11, and human PDIA1, showing signal peptide (SP), thioredoxin domains (Trx), and C-terminal ER retention signals. Essential, conserved cysteine residues are shown for each of the proteins' Trx domains. **B)** Regulation of protein expression using the *glmS* ribozyme system. The mRNA of interest encodes the ribozyme in the 3'UTR. Upon addition of glucosamine (GlcN, orange hexagon), which is converted to glucosamine-6-phosphate (pink hexagon) by the parasite, the ribozyme is activated to cleave the mRNA, leading to transcript instability and degradation (Prommana et al., 2013) **C)** Schematic of CRISPR/Cas9 mediated introduction of the *glmS* knockdown system into the genome. A repair template was transfected, along with a plasmid to express Cas9 and a gRNA, to introduce sequences for a 3xV5 tag, ER retention signals, stop codon, and *glmS* ribozyme. **D)** PCR integration test confirming correct modification of *pfpdi8* and *pfpdi11*. Correct integration results in increased amplicon size due to the V5 and *glmS* sequences. **E)** Western blots showing V5-tagged proteins in the *Pf*J2^apt-PDI8^glmS and *Pf*J2^apt-PDI11^glmS parasite lines at the predicted sizes for *Pf*PDI8 and -11. For each western blot, one of two biological replicates shown.

parasites [12,17]. In order to validate interaction between these PDIs and *Pf*J2, and to understand their roles in the *P. falciparum* ER, we used the *glmS* ribozyme to create conditional knockdown parasite lines for each protein in the background of *Pf*J2$^{apt}$ parasites [31] (Fig 5B). Using CRISPR/Cas9 genome editing, we introduced sequences for a 3xV5 tag and the *glmS* ribozyme into the *pfpdi8* or the *pfpdi11* locus (*Pf*J2$^{apt}$-PDI8$^{glmS}$ and *Pf*J2$^{apt}$-PDI11$^{glmS}$, respectively) (Fig 5C). Correct modifications of the loci were validated by PCR integration test, and V5-tagged proteins were visualized by western blot (Fig 5D and 5E).

The subcellular localizations of *Pf*PDI8 and -11 were determined by IFA, and both proteins were found to co-localize with *Pf*J2 in the ER (Fig 6A). To test the functionality of the *glmS* ribozyme knockdown system, each parasite line was treated with glucosamine (GlcN), and samples were taken for western blot analysis over the course of the parasite lifecycle. Compared to -GlcN control samples, protein levels were found to be reduced during GlcN treatment (S3 Fig). In order to determine the effect that PDI knockdown had on parasite growth, each cell line was treated with GlcN and parasite growth was measured over the course of two life cycles. We observed dramatic inhibition of parasite growth during *Pf*PDI8 knockdown, but no growth defects were observed when *Pf*PDI11 was knocked down (Fig 6B). These results demonstrate that *Pf*PDI8 is essential for the asexual lifecycle and suggest that *Pf*PDI11 may be dispensable, though in the case of *Pf*PDI11, the lack of a phenotype could be attributed to incomplete knockdown (S3 Fig). Both conclusions are supported by a genome-wide essentiality screen performed in *P. falciparum* [26].

*Pf*PDI8 contains two thioredoxin domains with classical CXXC active site cysteines, allowing the protein to function in oxidative folding as an oxidoreductase/isomerase [12,17]. *Pf*PDI11 has two thioredoxin domains containing noncanonical CXXS active sites, but likely maintains the ability to form mixed disulfide bonds with client proteins through the conserved cysteine residues [32–34]. To determine whether we could trap redox interactions between these PDIs and their substrates, *Pf*J2$^{apt}$-PDI8$^{glmS}$ and *Pf*J2$^{apt}$-PDI11$^{glmS}$ cultures were treated with DVSF, and protein lysates were collected for western blot analysis. Several high molecular weight bands containing *Pf*PDI8 appear following DVSF treatment, indicating that multiple substrates rely on the oxidoreductase activity of *Pf*PDI8, in contrast to *Pf*J2, whose western blot shows a narrower set of redox substrates (Fig 6C). Similar results were observed for PfPDI11 (Fig 6C).

We next sought to determine if DVSF crosslinking occurs through Trx-domain cysteines. To do this, we attempted to generate parasites overexpressing either wild-type copies of *Pf*J2, *Pf*PDI8, *Pf*PDI11, or overexpressing these proteins with cysteine-to-alanine mutations in the Trx domain active sites. These types of mutations abolish DVSF-crosslinking in Trx proteins of model organisms [28,30]. We were unable to generate parasites overexpressing wild-type or mutant copies of *Pf*J2. We were also unable to generate parasites overexpressing a mutant version of *Pf*PDI8, but were successful in creating parasites overexpressing the wild-type protein; characterization of that parasite line revealed mislocalization of the overexpressed *Pf*PDI8 (S4 Fig). Given the essential nature of *Pf*J2 and *Pf*PDI8, we concluded that the parasites may be sensitive to their overexpression and to mutations in their Trx domains.

In contrast, we were successfully able to generate both wild-type and cysteine-to-alanine *Pf*PDI11 overexpression mutants (*Pf*PDI11$^{wt}$ and *Pf*PDI11$^{mut}$, respectively) (S6 Fig). Both parasite lines displayed the expected ER co-localization with PfJ2 (S5 Fig). Importantly, treatment of these parasites with DVSF revealed extensive crosslinking between *Pf*PDI11 and substrates in the wild-type parasites, but crosslinking is abolished in parasites with cysteine-to-alanine mutations in the Trx domain (S5 Fig). These data demonstrated the specificity of DVSF for trapping redox partnerships in *P. falciparum*.

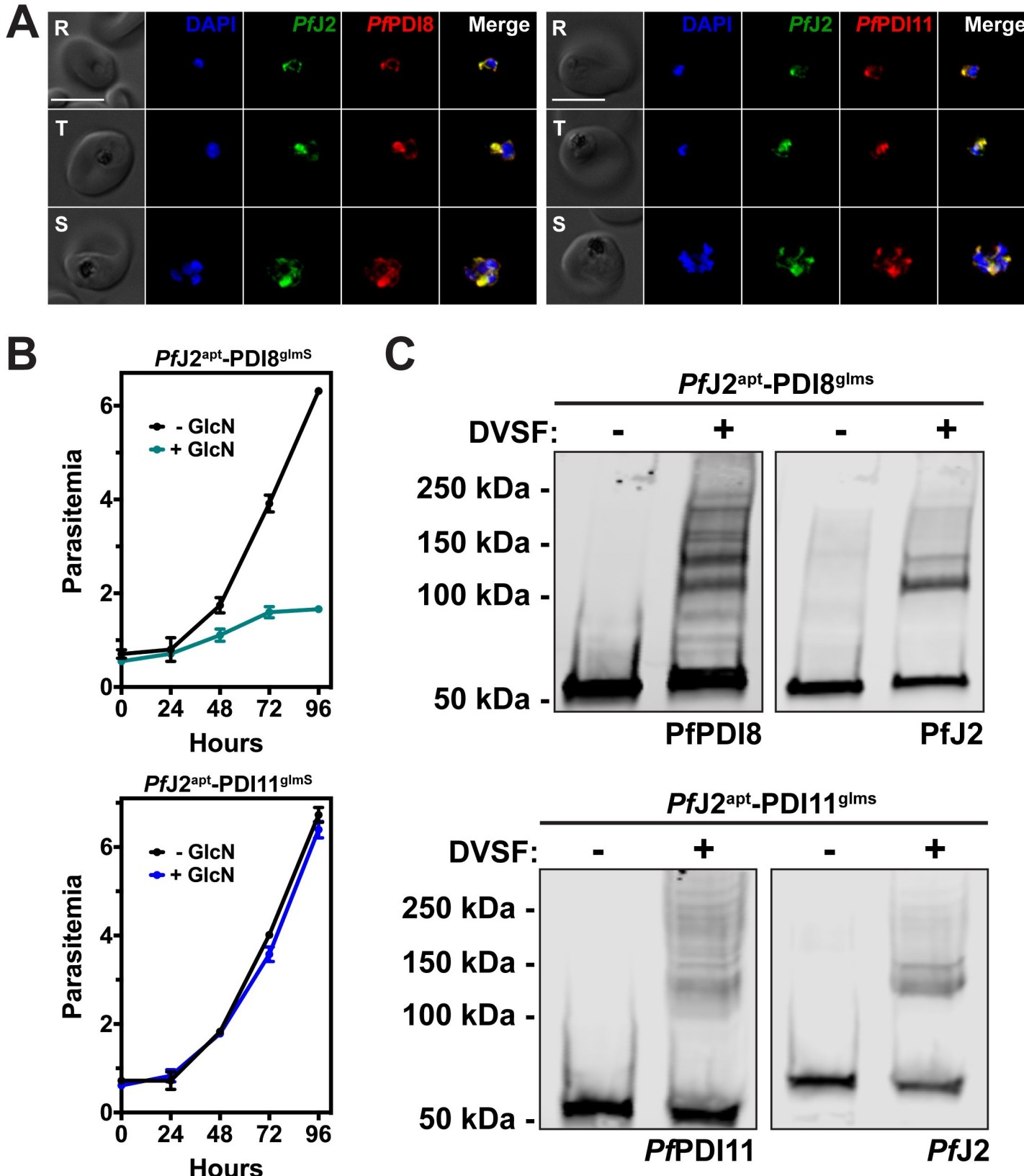

**Fig 6. *Pf*PDI8 and *Pf*PDI11 are redox-active ER proteins. A)** *Pf*J2$^{apt}$-PDI8$^{glmS}$ and *Pf*J2$^{apt}$-PDI11$^{glmS}$ parasites were fixed and stained with DAPI (blue) and with antibodies against HA (green) and V5 (red). Ring (R), Trophozoite (T), and Schizont (S) stage parasites are shown. Z-stack Images were deconvoluted and shown as a single, maximum intensity projection. Scale bar represents 5 μm. **B)** Asynchronous *Pf*J2$^{apt}$-PDI8$^{glmS}$ (top) and *Pf*J2$^{apt}$-PDI11$^{glmS}$ (bottom) parasites were grown in normal (-GlcN) or knockdown (+ 5mM GlcN) conditions, and parasite growth was monitored daily for 96 hours via flow cytometry. Data points represent the mean of three technical replicates, with error bars representing standard deviation. For each growth curve, a representative experiment of two biological replicates is shown. **C)** *Pf*J2$^{apt}$-PDI8$^{glmS}$ and *Pf*J2$^{apt}$-PDI11$^{glmS}$ parasites were incubated with 3 mM DVSF in 1x PBS for 30 minutes at 37˚C, then samples were taken for western blot analysis. Membranes were incubated with antibodies against HA (*Pf*J2) and V5 (*Pf*PDI8 or -11). For each western blow, one of two biological replicates is shown.

Given the unusual nature of the *Pf*PDI11 CXXS Trx-domain active site, we took advantage of these overexpression parasites to further investigate *Pf*PDI11 function. Trx-domain-proteins with CXXS active sites are largely under-studied in all organisms [32–34]. It is possible that the remaining active-site cysteines in *Pf*PDI11 forms mixed disulfides that may serve to retain proteins the in ER, prevent their aggregation, and/or block cysteines from non-productive bond formation as they fold [32,34]. Therefore, we hypothesized that we would be able to detect the mixed *Pf*PDI11-substrate disulfide bonds by non-reducing SDS-PAGE and western blotting. Indeed, we were able to detect high-molecular-weight species of *Pf*PDI11 when *Pf*PDI11$^{wt}$ parasite lysates were used for western blotting under non-reducing conditions (S6 Fig). In contrast, these species were missing when *Pf*PDI11$^{mut}$ parasite lysates were used (S6 Fig).

## The *Pf*BiP-*Pf*J2-*Pf*PDI8 oxidative folding complex

Having shown that *Pf*J2 and *Pf*PDI8 are redox partners and that both proteins are essential for the *P. falciparum* asexual lifecycle, we decided to focus on their interaction and what roles they may play together in the ER. To confirm the interaction between *Pf*J2 and *Pf*PDI8, *Pf*J2 was immunoprecipitated from *Pf*J2$^{apt}$-PDI8$^{glmS}$ parasites, and *Pf*PDI8 was found to co-immunoprecipitate (Fig 7A). When performing the reciprocal co-IP, we were unable to detect *Pf*J2 pulling down with *Pf*PDI8, perhaps due to inefficiency of the anti-V5 IP (S7 Fig). However, we were able to detect a band of overlapping *Pf*J2/*Pf*PDI8 signal when the *Pf*PDI8 IP was performed on cultures treated with DVSF, showing that the two proteins are interacting, redox partners (Fig 7B).

Our observations using the redox crosslinker DVSF showed that *Pf*PDI8 is a major redox partner for *Pf*J2, whereas *Pf*PDI8 has multiple other redox partnerships (Figs 4C and 6D). One explanation for this observation is that *Pf*J2 may work upstream to prime *Pf*PDI8 for interaction with its substrates. Therefore, we asked whether we could detect *Pf*PDI8+substrates co-immunoprecipitating with *Pf*J2. When *Pf*J2 was immunoprecipitated from *Pf*J2$^{apt}$-PDI8$^{glmS}$ parasites treated with DVSF, we were able to detect *Pf*PDI8 trapped to other substrates, indicated by smearing of the *Pf*PDI8 signal above 150 kDa (Fig 7C). As a negative control, *Pf*PMV was probed for in the proteins immunoprecipitating with *Pf*J2 and *Pf*PDI8 and was not found to interact with either protein, confirming the specificity of the anti-HA and anti-V5 immunoprecipitations utilized (S8 Fig). Together, these results confirm the interaction between *Pf*J2 and *Pf*PDI8 and suggest that *Pf*J2 may be part of a complex including *Pf*PDI8 and its substrates.

Oxidative folding in the ER, mediated by proteins such as *Pf*J2 and *Pf*PDI8, is only one aspect of protein folding in the ER, and likely works in conjunction with other folding determinants, such as the Hsp70 BiP. *Pf*J2 is an ER Hsp40—a class of co-chaperones that interact with the Hsp70 BiP—and BiP is likely involved in the folding of the same substrates that interact with *Pf*PDI8. Such cooperation between BiP and mediators of oxidative folding has received limited investigation in most organisms, to our knowledge. Therefore, we next asked whether *Pf*BiP interacts with *Pf*J2 and/or *Pf*PDI8.

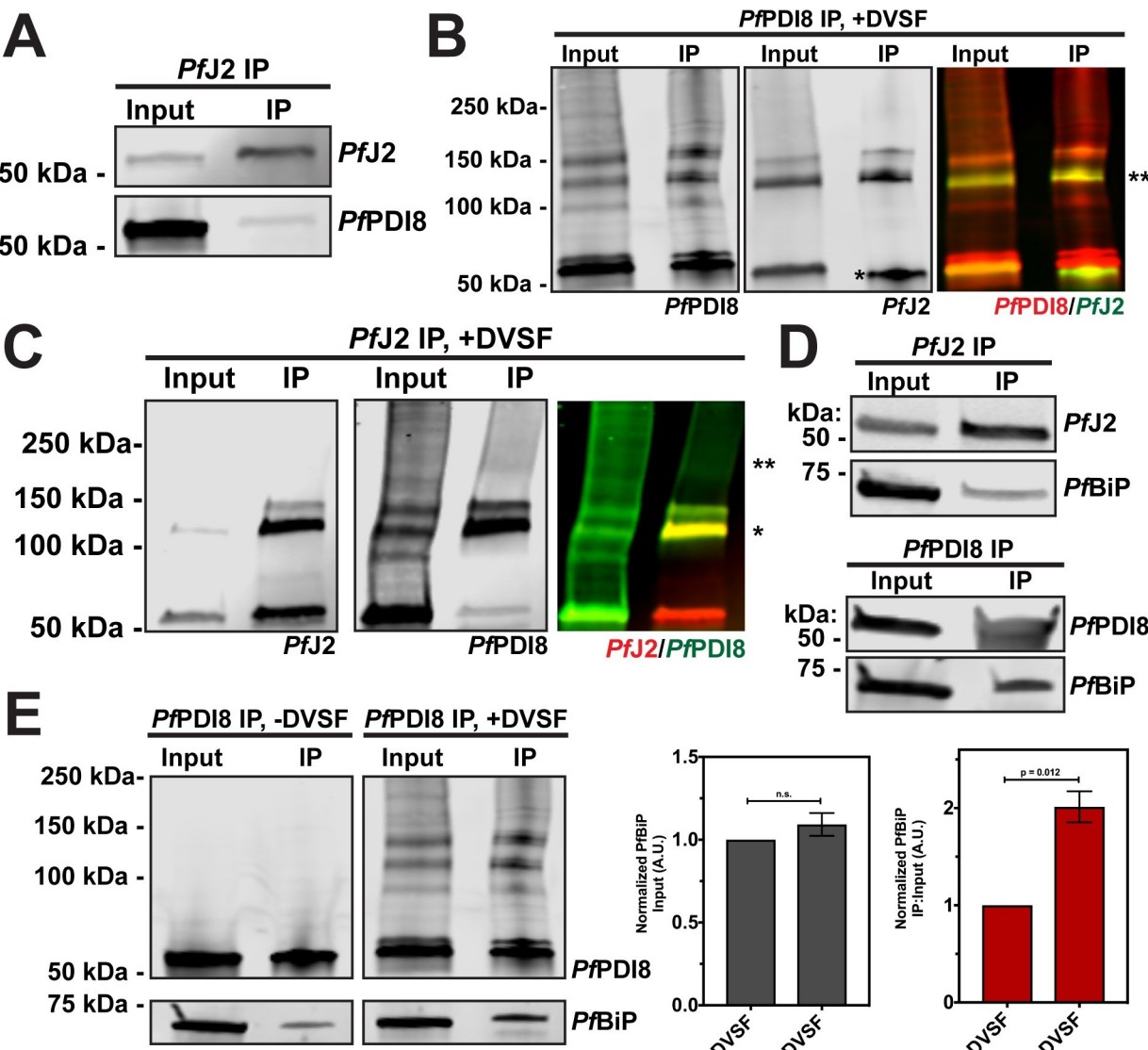

**Fig 7. The *Pf*BiP-*Pf*J2-*Pf*PDI8 oxidative folding complex. A)** *Pf*J2 and interacting proteins were immunoprecipitated from *Pf*J2$^{apt}$-PDI8$^{glmS}$ parasite lysate using anti-HA antibodies. Input and eluted IP samples were used for western blot analysis. Membrane was probed with HA and V5 antibodies to detect *Pf*J2 and *Pf*PDI8, respectively. **B)** *Pf*J2$^{apt}$-PDI8$^{glmS}$ parasites were incubated with 3 mM DVSF as described above, then V5 antibodies were used to immunoprecipitate *Pf*PDI8 and interacting proteins. Input and eluted IP samples were used for western blot analysis. Membrane was probed with V5 and HA antibodies to detect *Pf*PDI8 and *Pf*J2, respectively. Antibody heavy chain is indicated by the single asterisk (*) in the *Pf*J2 panel. A merged image of the *Pf*PDI8 (red) and *Pf*J2 (green) signal is shown, with the yellow overlap in signal indicated by a double asterisk (**). **C)** *Pf*J2$^{apt}$-PDI8$^{glmS}$ parasites were incubated with 3 mM DVSF as described above, then HA antibodies were used to immunoprecipitate *Pf*J2 and interacting proteins. Input and eluted IP samples were used for western blot analysis. Membrane was probed with HA and V5 antibodies to detect *Pf*J2 and *Pf*PDI8, respectively. A merged image of the *Pf*J2 (red) and *Pf*PDI8 (green) signal is shown, with a single asterisk (*) indicating the yellow overlap in signal and a double asterisk (**) indicating *Pf*PDI8+subtrates that co-immunoprecipitated with *Pf*J2. **D)** *Pf*J2 and *Pf*PDI8 were immunoprecipitated from *Pf*J2$^{apt}$ and *Pf*J2$^{apt}$-PDI8$^{glmS}$ parasite lysates, respectively. Input samples and eluted IP proteins were used for western blot analysis. Membrane was probed with HA and *Pf*BiP antibodies (top) or V5 and *Pf*BiP antibodies (bottom). Immunoprecipitations were performed in biological duplicate or triplicate, and representative results are shown in A-D. **E)** *Pf*J2$^{apt}$-PDI8$^{glmS}$ parasite cultures were evenly split into two conditions: 3 mM DVSF or PBS only for 30 minutes at 37°C, after which parasite lysates were used for anti-V5 immunoprecipitation. Input and eluted IP proteins were analyzed by western blot using V5 and *Pf*BiP antibodies. The *Pf*BiP signal was measured for each lane and the ratio of IP-to-Input signal was determined. N = 3 biological replicates. Error bars represent standard deviation.

Western blot analysis of proteins co-immunoprecipitating with *Pf*J2 revealed that *Pf*BiP does interact with *Pf*J2, consistent with our *Pf*J2 co-IP experiments and the proteins' predicted chaperone/co-chaperone roles (Figs 3 and 7D, top). Lack of a suitable antibody precluded

reciprocal *Pf*BiP immunoprecipitation to probe for *Pf*J2. However, as a control we showed that *Pf*BiP is not detected when wild-type parasites (lacking HA-tagged *Pf*J2) are subjected to anti-HA immunoprecipitation, ruling out nonspecific binding during the co-IP experiment (S9 Fig).

Next, we found that when *Pf*PDI8 was immunoprecipitated, *Pf*BiP was detected (Fig 7D, bottom). We further reasoned that because the same substrates may rely on both *Pf*PDI8 and *Pf*BiP to achieve their native state, trapping the *Pf*PDI8-substrate interaction with DVSF may increase the amount of *Pf*BiP co-immunoprecipitating with *Pf*PDI8. To test this hypothesis, parasite cultures were equally split into +/- DVSF treated aliquots, *Pf*PDI8 was immunoprecipitated, and lysates probed for *Pf*PDI8 and *Pf*BiP. Consistent with our hypothesis, we detected a two-fold increase in the amount of *Pf*BiP that pulled down with *Pf*PDI8 crosslinked to its substrates, with no significant difference in the starting amount of *Pf*BiP detected in the sample prior to immunoprecipitation (Fig 6E). Together, these results suggest that *Pf*J2 and *Pf*PDI8 work together with the major ER folding chaperone *Pf*BiP to help substrates reach their native states.

### ER redox interactions are druggable

Our data have shown that *Pf*J2 and *Pf*PDI8, which participate in ER redox partnerships with each other as well as other substrates, are essential proteins in the *P. falciparum* asexual life-cycle. These proteins, through inhibition of their redox interactions, may represent unexploited targets for antimalarials. Further, our data show that DVSF can target these redox partners, suggesting that these proteins may be druggable. Indeed, consistent with the idea of targeting ER redox proteins in disease, high-throughput drug screens have identified potent inhibitors of human PDI in an effort to combat upregulation that is associated with some cancers and neurodegenerative diseases [35–38].

We tested four of these commercially available PDI inhibitors—16F16, LOC14, CCF642, and PACMA31—for activity against cultured asexual *P. falciparum* parasites. In contrast to their reported, highly potent activity against human cells, we observed a wide range of $IC_{50}$ values for *P. falciparum* (S10 Fig). The compound with the best anti-*Plasmodium* activity was 16F16, with an $IC_{50}$ value of approximately 4 μM (Fig 8A) [39].

16F16 inhibits human PDI function by covalently binding the cysteines of the Trx domain active sites, thereby blocking their ability to catalyze oxidative folding [35,36]. If 16F16 behaves similarly in *P. falciparum*, we reasoned that treatment of cultures with 16F16 prior to performing redox crosslinking with DVSF would prevent crosslinking from occurring, as both compounds rely on the same cysteine residues for their activity. Indeed, pre-treatment with increasing amounts of 16F16 significantly and reproducibly decreased the amount of crosslinked *Pf*J2 detected by western blot (Fig 8B). Our data indicate that DVSF treatment crosslinks *Pf*J2 to *Pf*PDI8 and *Pf*PDI11 (Fig 4). Therefore, the observed reduction in *Pf*J2 crosslinking likely occurs due to direct reaction of 16F16 with the *Pf*J2 Trx-domain active site and/or the *Pf*PDI8 and -11 active sites. Similar experiments showed that pre-treatment of cultures with 16F16 also blocked crosslinking of *Pf*PDI8 and -11 with their substrates, though to a lesser extent than what was observed for *Pf*J2 (Fig 8C and 8D). The quantification of the Western blot signals for all replicates is found in S3 Table. These data suggest that 16F16 inhibits the redox activity of *Pf*J2, *Pf*PDI8, and *Pf*PDI11 (Fig 8B, 8C and 8D) and it is likely that this compound inhibits other Trx-domain containing proteins in *P. falciparum*.

These data suggest that redox interactions within the *P. falciparum* ER, occurring between essential proteins like *Pf*J2 and *Pf*PDI8 and their substrates, are sensitive to small molecule inhibition. Additionally, the disparity in activity observed for the PDI inhibitors against

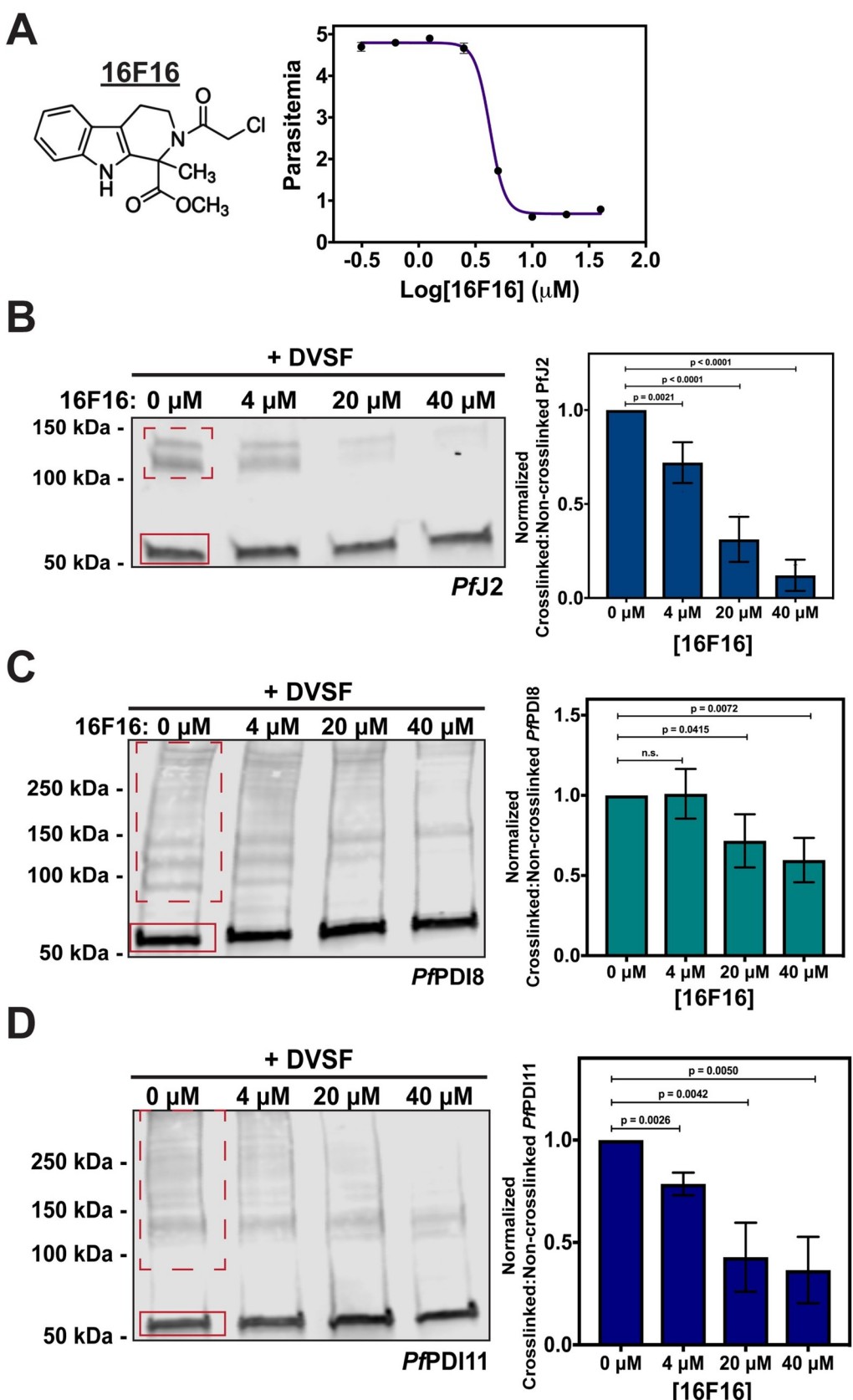

**Fig 8. ER redox interactions are sensitive to interruption by a small molecule. A)** Asynchronous $PfJ2^{apt}$ parasites were incubated in various concentrations of the human PDI inhibitor 16F16. Parasite growth was determined via flow cytometry at 72 hours and the 16F16 $IC_{50}$ was determined to be approximately 4 μM. Each data point in the curve represents the mean parasitemia at a given concentration, in technical triplicate. Error bars are not seen for data points in which they are smaller than the circle symbol, represent standard deviation from the mean. A representative $IC_{50}$ curve is shown for one of three biological replicates. **B)** $PfJ2^{apt}$, **C)** $PfJ2^{apt}$-PDI8$^{glmS}$, and **D)** $PfJ2^{apt}$-PDI11$^{glmS}$ parasites cultures were equally split and incubated with three concentrations of 16F16 for 3 hours prior to removal of 16F16, then incubation with 3 mM DVSF as described above. Samples were taken for western blot analysis, loading equal parasite equivalents into each gel. Membranes were incubated with antibodies against HA or V5. Signal for non-crosslinked (the band at approximately 50 kDa, solid red box) and crosslinked proteins (dashed red box) was measured. Inhibition was measured by determining the ratio of crosslinked to non-crosslinked signal. N = 3 biological replicates for each parasite line. Error bars represent standard deviation.

human and *P. falciparum* cell lines suggest that development of *Plasmodium*-specific inhibitors is likely possible (S10 Fig).

## Discussion

The ability to conduct oxidative folding likely underlies the diverse functions of the *P. falciparum* ER. The oxidizing environment of the ER encourages disulfide bond formation, but only the correct bonds allow proteins to reach their native states. Therefore, organisms must maintain a way to reduce/isomerize nonproductive disulfides. We have used CRISPR/Cas9 genome editing and conditional knockdown to show here that a putative disulfide reductase in the *P. falciparum* ER—*Pf*J2—Is essential for the parasite asexual lifecycle (Figs 1 and 2).

A co-IP/mass spectroscopy approach with stringent parameters for identifying interacting partners places *Pf*J2 in the broader context of ER biology, revealing that *Pf*J2 interacts with other folding determinants, such as BiP and Endoplasmin, as well as other members of the Thioredoxin superfamily, namely PDIs. The remaining proteins that were identified, most with unknown localization throughout the secretory pathway and many with no known function, may represent substrates that rely on *Pf*J2 and these other chaperones for their folding and/or trafficking. Among the proteins identified were large, complex proteins such as *Pf*MSP1 and *Pf*RhopH3 (Fig 3). Both proteins have numerous cysteine residues that must navigate oxidative folding when they are synthesized into the ER, likely relying on *Pf*J2 and *Pf*PDIs to do so correctly. Consistent with this hypothesis, a recent study identified *Pf*J2, *Pf*PDI11, and *Pf*Endoplasmin as potential contributors to folding and trafficking of *Pf*EMP1, a cysteine-rich transmembrane protein that serves as the major *P. falciparum* virulence factor [40]. Our analysis of the proteins identified by mass spectroscopy predicted interactions between *Pf*J2 and *Pf*BiP, and between *Pf*J2 and *Pf*PDI8. We confirmed these interactions via co-immunoprecipitation and western blot experiments. Notably, we also demonstrated an interaction between *Pf*J2 and *Pf*PDI11, a protein which was identified by mass spectroscopy (S1 Table) but did not meet our 5-fold enrichment criteria, suggesting that our parameters err on the side of caution to reduce false-positives.

One major *Pf*J2 redox substrate—*Pf*PDI8—was identified using a chemical biology approach. DVSF is a redox-specific crosslinker that has been used to identify redox partnerships between Thioredoxin proteins in the cytoplasm of model organisms [28–30]. To our knowledge, this compound had not yet been used to trap and define redox partnerships in *Plasmodium*, nor in the ER of any organism. We demonstrate its utility in the *Plasmodium* ER, using it to identify the redox partnership between *Pf*J2, *Pf*PDI8 and *Pf*PDI11 (Fig 4). Double conditional *P. falciparum* mutants showed that *Pf*PDI8 is also an essential, ER-resident protein and allowed us to probe more deeply the relationship between *Pf*J2 and *Pf*PDI8 (Figs 5,6 and 7). *Pf*J2 likely acts as a reductase, and previous *in vitro* characterization of *Pf*PDI8 revealed that

it behaves like a classical PDI, capable of both forming and reducing disulfide bonds [12,13,15–17].The propensity for *Pf*PDI8 to use its Trx domains either for oxidation of cysteines or reduction of disulfides likely depends on the oxidation state of its own active site (i.e. reduced *Pf*PDI8 can act as a reductase). One explanation for the redox partnership between *Pf*J2 and *Pf*PDI8 is that *Pf*J2 primes *Pf*PDI8 to act as a reductase for some or all of the numerous substrates we visualized using DVSF (Fig 6D). Consistent with this hypothesis, immunoprecipitation experiments showed that *Pf*PDI8+substrates pull down with *Pf*J2 (Fig 7C). We also found that *Pf*J2 and *Pf*PDI8 both interact with the Hsp70 *Pf*BiP, and detection of that interaction increases for *Pf*PDI8 when it is trapped to its substrates (Fig 7D and 7E). These data suggest a model in which *Pf*J2, *Pf*PDI8, and *Pf*BiP cooperate to ensure substrates in the ER correctly navigate the oxidative folding process to achieve their native states (Fig 9). As a predicted Hsp40 with an Hsp70-interacting J-domain, one of PfJ2's roles in this complex may be to recruit PfBiP to into a possible folding complex.

We also identified *Pf*PDI11 as a redox substrate of *Pf*J2 (Fig 4). Our data demonstrate that *Pf*PDI11 retains the ability to form mixed disulfides with client proteins despite the unusual CXXS Trx-domain active site (Figs 6, S5 and S6). Typically, the second cysteine of the Trx-domain active site is used to resolve enzyme-substrate mixed disulfides [10]. Therefore, the mechanisms used to resolve mixed disulfides between CXXS active sites and their substrates remains unclear, both in *P. falciparum* and other organisms. We propose that an ER-resident reductase such as *Pf*J2 helps resolve mixed disulfides, which would explain why *Pf*J2 and *Pf*PDI11 were found to be redox partners.

Collectively, our data also demonstrate the power and specificity of using DVSF for trapping redox interactions between proteins with Trx domains and their substrates in *P. falciparum*. We showed that DVSF does not crosslink *Pf*PMV or *Pf*BiP to other proteins, despite the presence of numerous cysteine residues in the former, and that using NEM to block sulfhydryl groups in the parasite prevents crosslinking between *Pf*J2 and its substrates (Figs 4 and S2). Furthermore, we showed that DVSF traps *Pf*PDI11 to its substrates, but cysteine-to-alanine mutations in the *Pf*PDI11 Trx domain active sites abolish this crosslinking even though this mutant has two cysteine residues that are not in the active site (S5 Fig). These data are consistent with observations in yeast and mammalian cells, in which DVSF has been validated to specifically trap Trx-domain proteins to the substrates with which they exchange disulfide bonds [28–30].

Importantly, given the recent stagnation observed in malaria elimination efforts, which is coincident with increasing cases of antimalarial resistance, we not only identified two proteins with essential functions; we further demonstrated that the redox partnerships of these proteins are sensitive to disruption by small molecule inhibition (Fig 8). 16F16 is a covalent inhibitor that blocks Trx-domain cysteines [35,36]. Such a compound, if specific for *P. falciparum*, could be expected to cripple oxidative folding in the ER and kill the parasite. Recently, interest in covalent inhibitors for treatment of human disease has renewed, with several covalent inhibitors approved for use by the United States Food and Drug Administration [41]. One particular concern with covalent inhibitors is the fact that mutagenesis of the target residue would result in resistance, but mutagenesis of Trx-domain cysteines would lead to loss of function in and of itself, presumably making this type of resistance harder to evolve.

Another concern is whether proteins with such conserved active sites and roles in biology would make appropriate drug targets. In reality, conserved proteins have given the field many of its validated and proposed drug targets. For example, the widely-used anti-malarial drug atovaquone targets Complex III of the mitochondrial electron transport chain [42]. Some other conserved proteins/complexes that have been proposed as anti-malarial drug targets include Cytochrome B, the TCP-1 Ring Complex chaperone, and the proteasome [43–46]. The

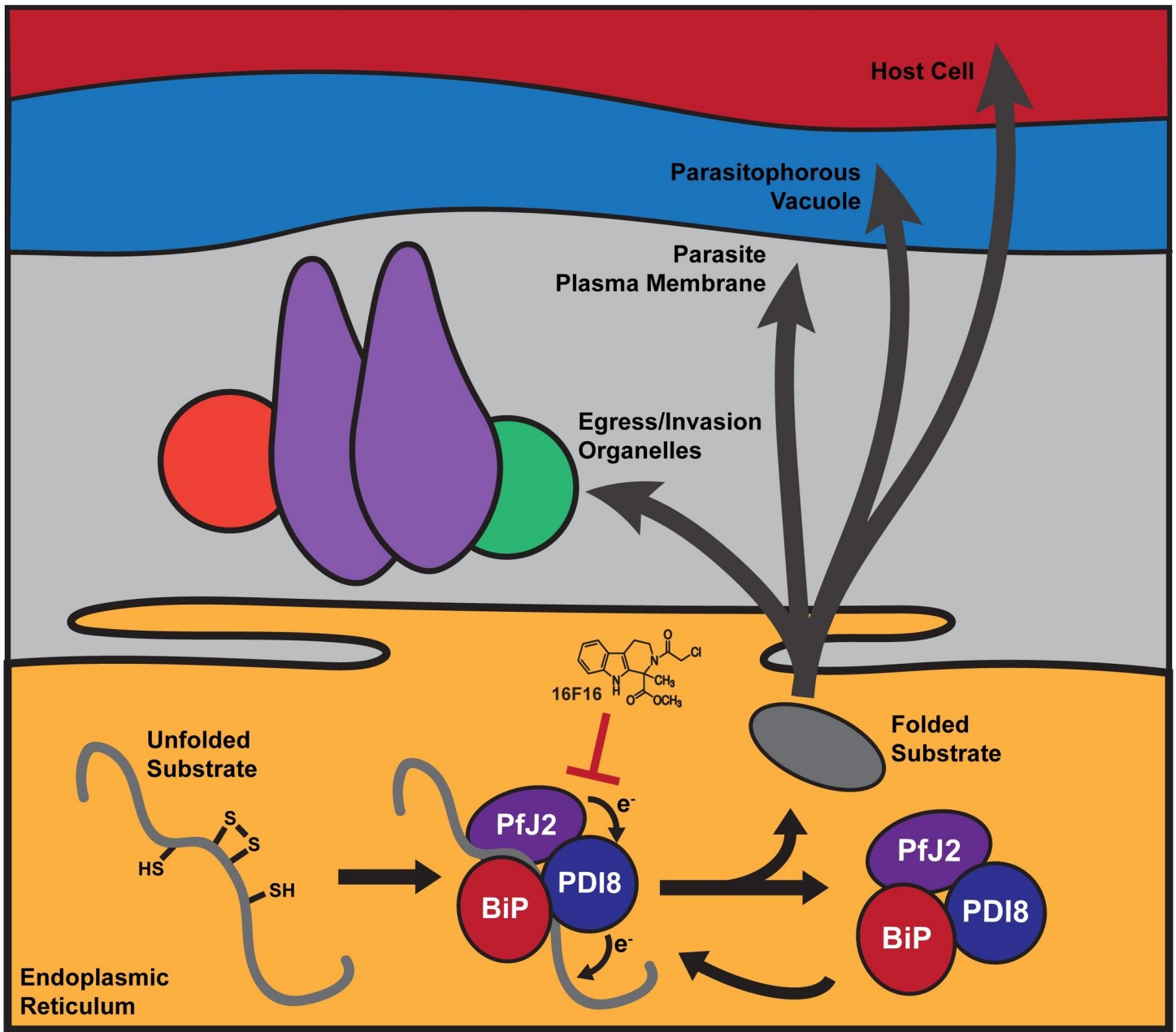

**Fig 9. Oxidative folding in the *P. falciparum* ER.** We propose that Trx-domain proteins like *Pf*J2 and *Pf*PDI8 work with *Pf*BiP to help nascent proteins, which perform essential functions within the ER and throughout the parasite secretory pathway, achieve their native states. The redox interactions between *Pf*J2, *Pf*PDI8, and their substrates are sensitive to inhibition by small molecules like 16F16, which could be expected to disrupt oxidative folding and impair the parasite's ability to perform functions essential for survival and replication.

precedence for targeting proteins that participate in conserved biology exists. Interest in targeting the *P. falciparum* proteasome began with observations that mammalian proteasome inhibitors have anti-malarial activity, which subsequently spurred development of *Plasmodium*-specific inhibitors [44,47,48]. Similarly, 16F16 likely targets many Trx-domains within the parasite, but we have used it to show that the redox interactions between *Pf*J2, *Pf*PDI8, and their substrates can be disrupted with a small molecule, and development of a *Plasmodium*-

specific inhibitor may be possible. In fact, given the disparity in activity observed for the PDI inhibitors against human cell lines and *P. falciparum*, enough diversity likely exists between these conserved proteins that *Plasmodium*-specific inhibitors could be developed (S10 Fig). Therefore, essential Trx-domain proteins in the parasite ER—like *Pf*J2 and *Pf*PDI8—represent a class of proteins and a pathway in the ER that is apt for antimalarial drug development.

## Materials and methods

### Construction of plasmids

Parasite genomic DNA was isolated from 3D7 parasites using QIAamp DNA blood kit (QIA-GEN). All constructs utilized in this study were confirmed by sequencing. Plasmids were constructed using the Sequence and Ligation Independent Cloning (SLIC) method. Plasmids to express Cas9 and gRNAs were constructed using pUF1-Cas9 as previously described [49–51]. All primers used in this study are listed in S4 Table. *pfpdi8* cDNA was prepared using TRIzol-extracted mRNA and reverse transcription with primer P20 (SuperScript III, Invitrogen). All restriction enzymes used in plasmid construction were purchased from New England Biolabs.

To generate pMG74-PfJ2, approximately 500 bp of the sequence encoding the *Pf*J2 C-terminus was amplified using primers P1 and P2, and approximately 500 bp from the pfj2 3'UTR were amplified using P3 and P4. The two amplicons were joined together via PCR sewing using P1 and P4, then inserted into pMG74 [19] digested with AflII and AatII. For expression of a *Pf*J2 gRNA, oligos P31 and P32 were inserted into pUF1-Cas9.

To generate pV5-glmS-PDI8, approximately 500 bp of the sequence encoding the *Pf*PDI8 C-terminus was amplified using primers P5 and P6. The 3x V5 tag was added to this amplicon via PCR sewing using a linearized plasmid encoding the 3xV5 sequence and primers P5 and P7. The glmS ribozyme sequence was amplified from pHA-glmS [31] using P8 and P9, then added to the *Pf*PDI8 C-terminus+V5 amplicon via PCR sewing using P5 and P9. The resulting amplicon was inserted into pHA-glmS that had been digested with AfeI and NheI, creating pPDI8-Cterm. Approximately 500 bp of the *pfpdi8* 3'UTR was amplified using P10 and P11, then inserted into pPDI8-Cterm that had been digested with HindIII and NotI, creating pV5-glmS-PDI8. For expression of a *Pf*PDI8 gRNA, oligos P33 and P34 were inserted into pUF1-Cas9.

To generate pV5-glmS-PDI11, approximately 500 bp of the sequence encoding the *Pf*PDI11 C-terminus was amplified using primers P12 and P13. The 3x V5 tag was added to this amplicon via PCR sewing using a linearized plasmid encoding the 3xV5 sequence and primers P12 and P14. The glmS ribozyme sequence was amplified from pHA-glmS [31] using 15 and P16, then added to the *Pf*PDI8 C-terminus+V5 amplicon via PCR sewing using P12 and P16. The resulting amplicon was inserted into pHA-glmS that had been digested with AfeI and NheI, creating pPDI11-Cterm. Approximately 500 bp of the *pfpdi11* 3'UTR was amplified using P17 and P18, then inserted into pPDI11-Cterm that had been digested with HindIII and NotI, creating pV5-glmS-PDI11. For expression of a *Pf*PDI11 gRNA, oligos P35 and P36 were inserted into pUF1-Cas9.

*Pf*PDI8 and *Pf*PDI11 overexpression was carried out by using CRISPR/Cas9 to insert the open reading frame (ORF) of the tagged genes into the *pfhsp110c* locus. pUC57-Hsp110, the repair plasmid targeting *pfhsp110*, includes the last 429 bp encoding the PfHsp110c (PF3D7_0708800) C-terminus, a 2A skip peptide sequence, sequences for various peptide tags, then the first 400 bp from the *pfhsp110c* 3'UTR. This plasmid was synthesized by GeneScript. For expression of a *Pf*Hsp110c gRNA, oligos P37 and P38 were inserted into pUF1-Cas9.

To generate pUC57-Hsp110-PDI8$^{wt}$, the *Pf*PDI8 ORF was amplified from cDNA using P19 and P20. A sequence encoding the 3xV5 was attached to this amplicon via PCR sewing using a

linearized plasmid encoding the tag and primers P19 and P21. The resulting amplicon was inserted into pUC57-Hsp110 digested with MfeI and SpeI.

To generate pUC57-Hsp110-PDI11$^{wt}$, the *Pf*PDI11 ORF was amplified using P22 and P23. A sequence encoding the 3xV5 was attached to this amplicon via PCR sewing using a linearized plasmid encoding the tag and primers P22 and P24. The resulting amplicon was inserted into pUC57-Hsp110 digested with MfeI and SpeI.

To generate pUC57-Hsp110-PDI11$^{mut}$, which required mutagenesis of the 2 Trx-domain active site, the PfPDI11 ORF was amplified in 3 parts. Part 1 was amplified using P25 and P26. Part 2 was amplified using P27 and P28. Part 3 was amplified using P29 and P30. Parts 1+2 were joined together using PCR sewing and primers P25 and P28. The resulting amplicon was attached to Part 3 using PCR sewing and primers P25 and P30. A sequence encoding the 3xV5 was attached to this amplicon via PCR sewing using a linearized plasmid encoding the tag and primers P25 and P24. The resulting amplicon was inserted into pUC57-Hsp110 digested with MfeI and SpeI.

## Parasite culture and transfection

*P. falciparum* asexual parasites were cultured in RPMI 1640 medium supplemented with Albu-MAX I (Gibco) and transfected as described earlier [52,53].

To generate the *Pf*J2$^{apt}$ parasite line, uninfected RBCs were transfected with 20 µg pMG74-PfJ2 (linearized prior to transfection using EcoRV) and 50 µg pUF1-Cas9-PfJ2, then fed to 3D7 parasites. Drug pressure was applied 48 hours after transfection, selecting for integration using 0.5 µM aTc and 2.5 µg/mL Blasticidin. After parasites grew back up from transfection and were cloned using limiting dilution, clones were maintained in medium containing 10 nM aTc and 2.5 µg/mL Blasticidin. Unless started otherwise, all +/- aTc growth experiments were conducted in medium containing 10 nM aTc and 2.5 µg/mL Blasticidin or medium containing only 2.5 µg/mL Blasticidin.

To generate the *Pf*J2$^{apt}$-PDI8$^{glms}$ parasite line, RBCs were transfected with 50 µg pV5-glmS-PDI8 and 50 µg pUF1-Cas9-PDI8, then fed to *Pf*J2$^{apt}$ parasites. Drug pressure was applied 48 hours after transfection, selecting with 0.5 µM aTc, 2.5 µg/mL Blasticidin, and 1 µM 5-Methyl[1,2,4]triazolo[1,5-a]pyrimidin-7-yl)naphthalen-2-ylamine (DSM1) [50]. After parasites grew back up from transfection and were cloned using limiting dilution, clones were maintained in medium containing 50 nM aTc and 2.5 µg/mL Blasticidin. *Pf*J2$^{apt}$-PDI11$^{glms}$ parasites were generated in the same manner, using 50 µg pV5-glmS-PDI11 and 50 µg pUF1-Cas9-PDI11.

To generate the *Pf*PDI8$^{wt}$ overexpression parasite line, RBCs were transfected with 50 µg pUC57-Hsp110-PDI8$^{wt}$ and 50 µg pUF1-Cas9-Hsp110, then fed to *Pf*J2$^{apt}$ parasites. Drug pressure was applied 48 hours after transfection, selecting with 0.5 µM aTc, 2.5 µg/mL Blasticidin, and 1 µM Drug Selectable Marker 1 (DSM1) [50]. After parasites grew back up from transfection and were cloned using limiting dilution, clones were maintained in medium containing 10 nM aTc and 2.5 µg/mL Blasticidin. *Pf*PDI11$^{wt}$ and *Pf*PDI11$^{mut}$ parasites were generated in the same manner, using pUC57-Hsp110-PDI11$^{wt}$ and pUC57-Hsp110-PD11$^{mut}$, respectively.

Parasite synchronization was carried out as described [54].

## Western blotting

Western blots were performed as previously described [55]. Briefly, ice-cold 0.04% saponin in 1x PBS was used to isolate parasites from host cells. Parasite pellets were subsequently solubilized in protein loading dye to which Beta-mercaptoethanol had been added (LI-COR

Biosciences) and used for SDS-PAGE. Primary antibodies used in this study were rat-anti-HA 3F10 (Roche, 1:3000), mouse-anti-HA 6E2 (Cell Signaling Technology, 1:1000), rabbit-anti-HA 715500 (Thermofisher, 1:100), mouse-anti-V5 TCM5 (eBioscence, 1:1000), rabbit-anti-V5 D3H8Q (Cell Signaling Technology, 1:1000), rabbit anti-*Pf*BiP MRA-1246 (BEI resources, 1:500), rabbit-anti-*Pf*EF1α (from D. Goldberg, 1:2000), and mouse-anti-*Pf*PMV (from D. Goldberg 1:400). Secondary antibodies used were IRDye 680CW goat-anti-rabbit IgG and IRDye 800CW goat-anti-mouse IgG (Li-COR Biosciences, 1:20,000). Membranes were imaged using the Odyssey Clx Li-COR infrared imaging system (Li-COR Biosciences). Images of membranes were processed using ImageStudio, the Odyssey Clx Li-COR infrared imaging system software (Li-COR Biosciences). Densitometry analysis of western blot signal was also performed using ImageStudio (Li-COR Biosciences).

## Microscopy and image analysis

Parasites were fixed for IFA using 4% Paraformaldehyde and 0.03% glutaraldehyde, then permeabilized with 0.1% Triton-X100. Primary antibodies used were rat-anti-HA 3F10 (Roche, 1:100), mouse-anti-HA 6E2 (Cell Signaling Technology, 1:100), mouse-anti-V5 TCM5 (eBioscence, 1:100), rabbit-anti-V5 D3H8Q (Cell Signaling Technology, 1:100), and mouse-anti-*Pf*PMV (from D. Goldberg 1:1). Secondary antibodies used were Alexa Fluor 488 and Alexa Fluor 546 (Life Technologies, 1:100). Cells were mounted to slides using ProLong Diamond with DAPI (Invitrogen). Fixed and stained cells were imaged using a DeltaVision II microscope system with an Olympus IX-71 inverted microscope. Images were collected as a Z-stack and deconvolved using SoftWorx, then displayed as a maximum intensity projection. Images were processed using Adobe Photoshop, with adjustments made to brightness and contrast for display purposes.

For imaging of parasite cultures using light microscopy, aliquots of culture were smeared onto glass slides and field-stained using Hema3 Fixative and Solutions (Fisher Healthcare), which is comparable to Wright-Giemsa staining. Slides were imaged using a Nikon Eclipse E400 microscope with a Nikon DS-L1-5M imaging camera. To measure parasite size, images were taken and parasites measured using ImageJ (NIH).

## Growth assays using flow cytometry

Aliquots of parasites culture were incubated in 8 μM Hoescht 33342 (Thermofisher Scientific) for 20 minutes at room temperature, then fluorescence was measured using a CytoFlex S (Beckman Coulter, Hialeah, Florida). Flow cytometry data were analyzed using FlowJo software(Treestar, Inc., Ashland, Oregon). For $IC_{50}$ experiments, data were analyzed using the 4-parameter dose-response-curve function of Prism (GraphPad Software, Inc.).

## Immunoprecipitation assays

Anti-HA immunoprecipitation (IP) assays were performed as previously described, using anti-HA magnetic beads (Pierce) [56]. Anti-V5 IP assays were performed in the same manner as with anti-HA, but anti-V5 magnetic beads were used according to manufacturer instructions (MBL International Corporation).

## Mass spectrometry and data analysis

CoIP samples were sent to Emory University Integrated Proteomics Core and analyzed using a Fusion Orbitrap mass spectrometer, or to the proteomics core at the Fred Hutchinson Cancer Research Center, where samples were analyzed using an OrbiTrap Elite. Data were searched

using Proteome Discoverer 2.2 with UP000001450 *Plasmodium falciparum* (Uniprot Nov 2018) as the background database. The validation also included Sequest HT and Percolator to search for common contaminants. Results consisted of high confidence data with a 1% false discovery rate. Protein abundance was calculated by summing the total intensities (MS1 values) of all matched peptides for each selected protein, and normalizing by the total summed intensity of all matched peptides in the sample, as previously described [20]. The resulted calculated value was then averaged between replicates. To filter for high confidence hits, we calculated the abundance ratio between *Pf*J2 IP and the parental control, and set a threshold of fivefold enrichment over the parental control. This list was further curated to exclude proteins without a signal peptide or a transmembrane domain to identify biologically relevant interactors, because proteins with an signal peptide and/or transmembrane domain potentially traverse through the ER and can come in contact with PfJ2 in this cellular context.

## Identification of PfJ2 redox partners

*Pf*J2[apt] parasites were incubated with 3 mM divinyl sulfone (DVSF, Fisher Scientific) in 1x PBS for 30 minutes at 37˚C, then used for anti-HA immunoprecipitation as described above. Immunoprecipitated proteins were separated by SDS-PAGE. Polyacrylamide gel slices corresponding to the protein molecular weights of interest were excised and the peptides extracted by in-gel enzymatic digestion. The gel slices were dehydrated in 100% acetonitrile and dried using a speed vac. The proteins were then reduced by rehydrating the gel slices in 10mM dithiothreitol in 100mM ammonium bicarbonate solution, and alkylated in 50mM iodoacetamide in 100 mM ammonium bicarbonate solution. The gel slices were then washed in 50% acetonitrile in 50mM ammonium bicarbonate solution before being dehydrated and dried again using 100% acetonitrile and a speed vac. Proteins were then digested in-gel by rehydrating the gel slices in trypsin enzyme solution consisting of 6ng/µl trypsin (Promega) in 50mM ammonium bicarbonate solution. Digestion was performed at 37˚C overnight. Peptides were extracted through stepwise incubations with 2% acetonitrile and 1% formic acid solution, 60% acetonitrile and 0.5% formic acid solution, and 100% acetonitrile solution. Supernatants were combined and dried in a speed vac before resuspension in 20 µl water with 0.1% formic acid.

LC-MS/MS analysis was performed using a 50 cm fused silica capillary (75 µm ID) packed with C18 (2 µm, Dr. Maisch GmbH), and heated to 50˚C. Prior to loading the column, sample was loaded onto a 2 cm Acclaim PepMap 100 (Thermo Fisher Scientific) trap (75 µm ID, C18 3 µm). For each sample injection, 5 µl of sample was loaded onto the trap using an Easy nLC-1000 (Thermo Fisher Scientific). Each sample was separated using the Easy nLC-1000 with a binary mobile phase gradient to elute the peptides. Mobile phase A consisted of 0.1% formic acid in water, and mobile phase B consisted of 0.1% formic acid in acetonitrile. The gradient program consisted of three steps at a flow rate of 0.3 µL/min: (1) a linear gradient from 5% to 40% mobile phase B over two hours, (2) a 10 minute column wash at 80% mobile phase B, and (3) column re-equilibration for 20 minutes at 5% mobile phase B.

Mass spectra were acquired on a Fusion Lumos Tribrid (Thermo Fisher Scientific) mass spectrometer operated by data dependent acquisition (DDA) using a top 15 selection count. Precursor ion scans were performed at 120,000 resolution over a range from 375 to 1375 m/z. DDA was performed with charge exclusion of 1 and greater than 8, with isotope exclusion, and dynamic exclusion set to 10 seconds. MS/MS was performed using an isolation window of 1.6 m/z for selection, normalized collision energy (NCE) of 28, and higher energy collision induced dissociation (HCD). MS/MS spectra were acquired at 15000 resolution with an automatic gain control (AGC) target of 70,000 and maximum injection time of 50 ms.

Mass spectra (.raw files) were converted to mzML format using MSConvert (version 3.0.1908) [57] and peptide sequences were identified using database searching with Comet [58] (version 2016.01 rev 2). Spectra were searched against a subset of *P. falciparum* secretory proteins, common laboratory contaminants, and an equal number of randomized decoy sequences (4386 total protein sequence). Comet parameters included variable modifications of +57.021464 Da or +118.0089 Da on cysteine and a variable modification of +15.994915 Da on methionine or tryptophan. Precursor mass tolerance was set to 25 ppm and a fragment_bin_-tolerance of 0.02 and fragment_bin_offset of 0 were used. Full-tryptic enzymatic cleavage was set, allowing for up to 3 missed cleavages. Peptide spectrum matches (PSM) were analyzed using the Trans-Proteomic Pipeline [59] (TPP, version 5.0.0 Typhoon), to assign peptide and protein probabilities using PeptideProphet [60] and iProphet [61], respectively. Spectral counts and precursor ion intensities were exported for each non-redundant PSM at a 1% false discovery rate (FDR). Protein inference was performed with ProteinProphet [62], using a 1% FDR.

## Supporting information

**S1 Table. *Pf*J2 co-immunoprecipitating proteins identified by tandem mass spectroscopy.** *Pf*J2 was immunoprecipitated from *Pf*J2apt parasite lysates using anti-HA antibodies, and co-immunoprecipitating proteins were identified by tandem mass spectrometry (MS/MS) analysis. Control parental parasites (lacking HA-tagged *Pf*J2) were also used for immunoprecipitation and analyzed in the same manner. Abundance of each identified protein was calculated by summing the total MS1 intensities (highlighted in yellow) of all matched peptides for each selected protein, and normalizing by the total summed intensity of all matched peptides in the sample.
(XLSX)

**S2 Table. Proteins crosslinked to PfJ2 by DVSF identified by tandem mass spectroscopy.** *Pf*J2apt cultures were treated with DVSF and immunoprecipitation of *Pf*J2 was performed. The immunoprecipitation was also performed in parallel using cultures that had not received DVSF treatment. The immunoprecipitated proteins were subjected to separation by SDS-PAGE, visualized using Coomassie, and sections of the gel were excised and used for MS/MS protein identification (See Fig 4).
(XLSX)

**S3 Table. Quantification of DVSF crosslinking inhibition by pre-treatment of parasites with 16F16.** Signal for non-crosslinked and crosslinked proteins was measured (See Fig 8). Inhibition was measured by determining the ratio of crosslinked to non-crosslinked signal. Data were normalized to the control sample which did not receive 16F16 treatment. N = 3 or 4 biological replicates for each parasite line.
(XLSX)

**S4 Table. Primers used in this study.** List of every primer and guide RNA used to generate the transfected plasmids as well as to determine the status of the targeted gene.
(XLSX)

**S1 Fig. Parasite development is slowed during PfJ2 knockdown.** *Pf*J2apt parasites were tightly synchronized (0–3 hours) to the ring stage, then split into either +aTc (10 nM) or–aTc medium. Smears were made and field-stained at various time points throughout the asexual lifecycle. Stained slides were imaged and parasite size was measured. Unpaired t-test, ****

indicates p $\leq$ 0.0001. Representative experiment of 3 biological replicates shown.
(TIF)

**S2 Fig. PfBiP SDS-PAGE migration is unaffected by DVSF.** $Pf$J2$^{apt}$-PDI8$^{glms}$ parasites were treated with 3 mM DVSF in 1xPBS for 30 minutes at 37°C, or left untreated as a control, and parasite lysates were used for western blotting. Membranes were probed with antibodies against $Pf$BiP. Asterisks (*) denote nonspecific bands.
(TIF)

**S3 Fig. GlcN treatment leads to knockdown of $Pf$PDI8 and $Pf$PDI11.** Asynchronous $Pf$J2$^{apt}$-PDI8$^{glms}$ (top) and $Pf$J2$^{apt}$-PDI11$^{glms}$ (bottom) parasites were treated with 5 mM GlcN and samples were taken for western blot analysis at 0 (before addition of GlcN), 24, 48, and 72 hours. Membranes were probed with antibodies for V5 and $Pf$PMV.
(TIF)

**S4 Fig. Overexpression of $Pf$PDI8 results in mislocalization. A)** Top: schematic of exogenous V5-tagged, wild-type $Pf$PDI8 expression using the $pfhsp110$ (PF3D7_ 0708800) locus and a T2A skip peptide in the $Pf$PDI8$^{wt}$ parasite line, created in the background of $Pf$J2$^{apt}$ parasites. Bottom: western blot of parental $Pf$J2$^{apt}$ and $Pf$PDI8$^{wt}$ parasite lysates, probed for antibodies against the V5 tag ($Pf$PDI8) and the HA tag ($Pf$J2). **B)** $Pf$PDI8$^{wt}$ parasites were glutaraldehyde/paraformaldehyde fixed and used for IFA. Staining was carried out using DAPI (blue), antibodies against the HA tag ($Pf$J2, green), and the V5 tag ($Pf$PDI8, red). Scale bar represents 5 μm.
(TIF)

**S5 Fig. Characterization of PfPDI11 overexpression lines. A)** Left: schematic of exogenous V5-tagged, $Pf$PDI11 expression using the $pfhsp110$ (PF3D7_ 0708800) locus and a T2A skip peptide in the $Pf$PDI11$^{wt}$ and $Pf$PDI11$^{mut}$ parasite lines, created in the background of $Pf$J2$^{apt}$ parasites. In $Pf$PDI11$^{mut}$ parasites, both Trx-domain CXXS active sites were changed to AXXA. Right: western blot of $Pf$PDI11$^{wt}$ and $Pf$PDI11$^{mut}$ parasite lysates, probed for antibodies against the V5 tag ($Pf$PDI11) and the HA tag ($Pf$J2). **B)** $Pf$PDI11$^{wt}$ and $Pf$PDI11$^{mut}$ parasites were glutaraldehyde/paraformaldehyde fixed and used for IFA. Staining was carried out using DAPI (blue), antibodies against the HA tag ($Pf$J2, green), and the V5 tag ($Pf$PDI8, red). Scale bar represents 5 μm. **C)** $Pf$PDI11$^{wt}$ and $Pf$PDI11$^{mut}$ parasites were treated with 3 mM DVSF in 1x PBS for 30 minutes at 37°C, then parasite lysates used for western blotting. Membranes were probed with antibodies against the V5 tag ($Pf$PDI11) and the HA tag ($Pf$J2).
(TIF)

**S6 Fig. PfPDI11 non-reducing western blots.** Saponin-isolated $Pf$PDI11$^{wt}$ and $Pf$PDI11$^{mut}$ parasites were dissolved in protein loading dye lacking a reducing agent and used for western blotting. Membranes were probed with antibodies against the V5 tag ($Pf$PDI11) and the HA tag ($Pf$J2).
(TIF)

**S7 Fig. Immunoprecipitation of PfPDI8.** $Pf$PDI8 and interacting proteins were immunoprecipitated from $Pf$J2$^{apt}$-PDI8$^{glmS}$ parasite lysate using anti-V5 antibodies. The pre-IP input sample, the sample eluted from the antibodies (IP), and the sample removed from the beads containing the unbound proteins (Ub) were used for western blotting. Membranes were probed with antibodies against V5 and HA. The asterisk (*) denotes the heavy chain of the antibody used for immunoprecipitation.
(TIF)

**S8 Fig. Anti-HA, anti-V5 IP.** *Pf*J2$^{apt}$-PDI8$^{glmS}$ parasite lysates were used for either (A) anti-HA immunoprecipitation (IP) or (B) anti-V5 IP. The pre-IP input sample (In), the sample eluted from the antibodies (Elu), and the sample removed from the beads containing the unbound proteins (Ub) were used for western blotting. Membranes were probed with antibodies against HA to detect *Pf*J2, V5 to detect *Pf*PDI8, *Pf*PMV. The asterisk in (B) indicates the heavy chain of the anti-V5 antibody used for immunoprecipitation.
(TIF)

**S9 Fig. Anti-HA BiP IP.** 3D7 parasite lysates were used for anti-HA immunoprecipitation (IP). The pre-IP input sample (In), the sample eluted from the antibodies (Elu), and the sample removed from the beads containing the unbound proteins (Ub) were used for western blotting. Membranes were probed with antibodies against HA and *Pf*BiP.
(TIF)

**S10 Fig. PDI Inhibitor IC50s.** Left: Asynchronous 3D7 parasites were incubated in various concentrations of human PDI inhibitors. Each data point represents the mean parasitemia at a given concentration, in technical triplicate, at 72 hours. Error bars are not seen for data points in which they are smaller than the circle symbol, represent standard deviation from the mean. Representative IC$_{50}$ curves are shown for each drug. Experiments were performed in biological triplicates. Right: table describing each of the human PDI inhibitors used and their calculated *P. falciparum* IC$_{50}$ values.
(TIF)

## Acknowledgments

We thank Dan Goldberg for antibodies against *Pf*PMV and *Pf*EF1α; Julie Nelson at the CTEGD Cytometry Shared Resource Laboratory for help with flow cytometry and analysis; and Muthugapatti Kandasamy at the Biomedical Microscopy Core at the University of Georgia for help with microscopy. We acknowledge assistance of the Proteomics Resource at Fred Hutchinson Cancer Research Center and the Emory University Integrated Proteomics Core for mass spectrometry and data analysis.

## Author Contributions

**Conceptualization:** David W. Cobb, Vasant Muralidharan.

**Data curation:** Michael R. Hoopmann, Rodrigo P. Baptista.

**Formal analysis:** David W. Cobb, Michael R. Hoopmann, Rodrigo P. Baptista, Vasant Muralidharan.

**Funding acquisition:** Robert L. Moritz, Vasant Muralidharan.

**Investigation:** David W. Cobb, Heather M. Kudyba, Alejandra Villegas, Baylee Bruton, Michelle Krakowiak.

**Methodology:** David W. Cobb, Vasant Muralidharan.

**Project administration:** Vasant Muralidharan.

**Resources:** Michael R. Hoopmann, Robert L. Moritz.

**Software:** Michael R. Hoopmann, Rodrigo P. Baptista.

**Supervision:** Vasant Muralidharan.

**Validation:** David W. Cobb, Heather M. Kudyba, Alejandra Villegas, Baylee Bruton, Michelle Krakowiak.

**Visualization:** David W. Cobb, Vasant Muralidharan.

**Writing – original draft:** David W. Cobb, Vasant Muralidharan.

**Writing – review & editing:** David W. Cobb, Michael R. Hoopmann, Robert L. Moritz, Vasant Muralidharan.

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
