## [Decision Letter · Decision Letter 0]

24 Jul 2020

Dear Vasant,

Thank you very much for submitting your manuscript "A druggable oxidative folding pathway in the endoplasmic reticulum of human malaria parasites" for consideration at PLOS Pathogens. As with all papers reviewed by the journal, your manuscript was reviewed by members of the editorial board and by several independent reviewers. In light of the reviews (below this email), we would like to invite the resubmission of a significantly-revised version that takes into account the reviewers' comments.

As you will see, all 3 reviewers had a range of important concerns with the paper. All agreed that the functional significance of the diverse set of proteins identified in the pull-downs of PfJ2 was unclear, given the absence of further validation of many of the putative interactions, and that additional controls are essential for these experiments. Several related questions were raised over technical aspects of the immunoprecipitation experiments. Questions were also raised over the validity of DVSF as a selective, cysteine-reactive cross-linker (how well has its specificity been confirmed in previous work?) and the on-target selectivity of 16F16. Overall, there was a broad consensus that the conclusions of the work are overstated and not fully supported in some cases by the data. Citation of related previous work also requires attention.

We cannot make any decision about publication until we have seen the revised manuscript and your response to the reviewers' comments. Your revised manuscript is also likely to be sent to reviewers for further evaluation.

Sincerely,

Michael J Blackman

Associate Editor

PLOS Pathogens

Margaret Phillips

Section Editor

PLOS Pathogens

Kasturi Haldar

Editor-in-Chief

PLOS Pathogens

orcid.org/0000-0001-5065-158X

Michael Malim

Editor-in-Chief

PLOS Pathogens

orcid.org/0000-0002-7699-2064

As you will see, all 3 reviewers had a range of important concerns with the paper. All agreed that the functional significance of the diverse set of proteins identified in the pull-downs of PfJ2 was unclear, given the absence of further validation of many of the putative interactions, and that additional controls are essential for these experiments. Several related questions were raised over technical aspects of the immunoprecipitation experiments. Questions were also raised over the validity of DVSF as a selective, cysteine-reactive cross-linker (how well has its specificity been confirmed in previous work?) and the on-target selectivity of 16F16. Overall, there was a broad consensus that the conclusions of the work are overstated and not fully supported in some cases by the data. Citation of related previous work also requires attention.

Reviewer's Responses to Questions

**Part I - Summary**

Reviewer #1: Overall this is an interesting paper that characterizes several thioredoxin domain-containing proteins in P. falciparum. The experiments are interesting. However, the conclusions go significantly beyond the data in several places. Fundamentally it is a good paper but it should be written such that the conclusions are not overstated and are solidly supported by the data. If the authors are willing to do this it could be considered for PLOS pathogens. As the conclusions are currently overstated it is difficult to assess some aspects of the paper.

The authors initial focus is on the Trx domain-containing protein PfJ2. They make a tagged and conditional mutant of the PfJ2 encoding gene and show that the protein is ER-localised and important for normal blood stage replication. This is convincing.

The authors identify interacting partners of PfJ2 using immunoprecipitation. The authors provide a list of 23 proteins that are proposed to be specific interactors of PfJ2. The criteria for inclusion on the list includes presence of a signal sequence or TM domain and enrichment in the experiment over control. It is unclear how much confidence the reader should have in the majority of proteins on the list. If the signal sequence/TM criteria was not applied to the list, how many proteins would be similarly enriched in the experiment over control. The key question is do we have any confidence in the protein interactions that were not validated elsewhere. As the data is presented it is unclear how the reader should interpret this list - the data needs to be presented more clearly (also applies to S2).

The functional significance of this list of proteins is also unclear. The authors speculate that the list suggests that PfJ2 works in the ER with other chaperones and is important for folding of many proteins with diverse locations in the cell. This list of possible interactions does not really provide strong support this hypothesis. This would require a significant number of experiments to support this hypothesis; likely beyond the scope of a single paper.

The authors then describe a series of experiments using the crosslinker DVSF. The cross linker appears to preferentially react with the highly reactive cys residues in redox proteins but it has not been widely used elsewhere (the three references provided appear to be the only references - if there are more the authors should provide them but this appears to be a relatively uncharacterised reagent). The experiments are interesting and provide interesting data but the section needs to be re-written with consideration of the following.

The authors use cross linking to support various points;

The first point demonstrated using DVSF is 'PfJ2 is redox active' - the protein does contain a Trx domain so likely contains reactive cys residues. Does the fact that it can be cross linked to other proteins mean that the protein is redox active. What exactly does 'redox active' mean when defined in this way? What is the significance of this observation and the statement that the protein is 'redox active'.

The second point demonstrated using DVSF is 'We next sought to test the hypothesis that PfJ2 contributes to ER oxidative folding by determining whether its putative Trx domain is redox-active and capable of forming mixed disulfides with its clients. To this end we employed the bifunctional, electrophilic crosslinker divinyl sulfone (DVSF)...'

Crosslinking with DVSF indicates that two proteins can be cross linked with DVSF. It in no way shows that a protein is involved in oxidative folding or forms mixed disulfides. Likewise, it does not show whether a protein is a client of a thioredoxin domain protein or vice versa. If this is the conclusion that the authors wish to put forward they would need to do many more experiments to look at protein folding in the cell directly and assess the impact of mutants on the folding of a defined substrate.

The third point demonstrated using DVSF is: 'This specificity allows DVSF to covalently and irreversibly trap Trx-domain proteins to their redox substrates'

There is no indication that any of the cross linked proteins are substrates (as above). If anything, it seems likely that the cross linked proteins, such as PDI8, play a role in resolving the disulfides in PfJ2.

The experiments using DVSF are interesting but to demonstrate all of these points would take more data than could reasonably be crammed into one paper. The claims are quite bold and would require significantly more than cross linking with DVSF to demonstrate. However, the experiments do provide useful information. The authors need to state the aims and conclusions of each experiment in molecular terms that are clear, precise, accurate, and not overstated, and use these terms consistently throughout the paper. My major objection is the overstatement of the conclusions that are not supported by the data.

The authors show that the PfJ2 can be cross linked using DVSF to several other proteins including PfPDI8 and 11. The experiments knocking down the genes that encode these proteins are interesting and seem convincing. The recruitment of BiP to the complex after cross linking could be discussed more critically in the discussion.

In the experiments using the 16F16 inhibitor of PDI it should also be mentioned that many other thioredoxin domain containing proteins could also be targets of the drug in addition to those studied by the authors.

Other points:

Discussion - Generally, the conclusions are overstated. For example, the authors state 'Whether PfJ2 homologs in other eukaryotes recruit BiP in this fashion remains to be seen, but our data suggest that this role may be conserved in other eukaryotes'. Given that all of the experiments are done in plasmodium I don't really see how the authors come to this conclusion? There are other examples of overstated conclusions also.

Table 1: Fold change - probably should be fold-enrichment. 

Supplementary figure 7 - experiment aims to show formation of mixed disulfides. It would be preferable to show both the reduced and non-reduced gels.

MS tables - data columns are formatted differently between experiments making comparison difficult.

Reviewer #2: This manuscript by Cobb et al describes work characterizing redox-active proteins of the P. falciparum ER.

The authors used the TetR-PfDOZI aptamer system to generate conditional mutants of the redox-active ER chaperone, PfJ2, and showed that it is essential for parasite survival. This is anticipated given its low mutagenesis score from the Zhang et al largescale piggyBac insertion mutagenesis study.

The authors determined the cellular location as the ER. This is consistent with a previous study which the authors have failed to cite (Oehring et al 2012; https://genomebiology.biomedcentral.com/articles/10.1186/gb-2012-13-11-r108)

The authors used immunoprecipitation methods in an attempt to identify interacting proteins. The authors used a filter on the mass spec data. They state that they only considered proteins that were “present in all three PfJ2apt coIP experiments, and were at least 5-fold more abundant compared to the Controls” (i.e. no antibody). They employed an additional filter and only considered proteins containing a signal peptide and/or at least one transmembrane domain.

The IP identified a very large number of proteins which seems inconsistent with a successful pull-down of PfJ2 interacting proteins. That is, there appears to be a lot of non-specific precipitation, with many proteins that are known not to be ER-located ranked more highly in the numbers of peptides identified. Indeed, PDI-11 and PDI-8, proteins that the authors have singled out for further study, are ranked 1494th and 3350th in PfJ2 Rep 3 pull-down (Supp Table 1). It is also not clear exactly how the “5-fold more abundant” filter was applied. For example, the bait protein is identified as the 10th hit in the PfJ2 Rep 1 (with 33 peptides) and as the 38th ranked hit in the Control Rep 1 (with 28 peptides). This confusion may just arise from inadequately described methods and sub-optimal data presentation. As another example of issues with trying to assess the data in Supp Table I, the third replicate (PfJ2 Rep 3) is presented in a different format to the other two replicates.

One of the proteins listed in Table 1 is PF3D7_1324900 - L-lactate dehydrogenase. One would anticipate that this glycolytic enzyme would be located in the parasite cytoplasm.

The authors used an innovative redox-active cysteine crosslinker to identify its substrates, which are other mediators of folding in the oxidizing environment of the ER.

The authors show, using the GlmS system, that knockdown of PDI-8 is associated with inhibition of growth while knock-down of PDI-8 is not deleterious. As the authors note, these studies are consistent with the relative mutagenesis score from the Zhang et al largescale piggyBac insertion mutagenesis study.

The authors were unable to generate transgenic parasite lines overexpressing PfJ2 or PDI-8. By contrast, a previous study (Oehring et al 2012) did report did generation of transgenic parasite lines expressing full-length C-terminally tagged PfJ2 from an episomally maintained plasmid. The authors should comment.

Co-IP experiments were used to confirm the interactions (Fig 6). It would have been useful to include as a control a protein that is not expected to be immunoprecipitated (eg a cytoplasmic protein).

The authors examined four commercially available PDI inhibitors. Only the irreversible alkylating agent, 16F16, showed any activity; and then only at uM levels. At this concentration, 16F16 is likely to react with thiol groups in multiple proteins. The authors could determine whether the PDI knock-down strains are more susceptible to 16F16 as a means of determining its activity is on-target. Without such validation it is over ambitious to claim that PDIs are a plausible target for new antimalarials.

Minor points:

Line 288 - The authors state “.. PfPDI8 and -11 remain unstudied in vivo..”. It would be better to say “remain unstudied in cultured parasites”. “in vivo” implies that the studies involve infected animals.

Fig 2D. The images of the untreated culture suggest that the line has a very short life cycle. Rings are observed at the same stage a 0 h and 39 h. Is this a particular trait of this 3D7 line?

Reviewer #3: The manuscript “A druggable oxidative folding pathway in the endoplasmic reticulum of human malaria parasites” describes interesting and important work that merits publication pending major modifications.

In this work, Cobb and co-workers explore the role of ER oxidative folding in malaria. This in presented in context of a new avenue for drug discovery. Within this discussion, it would be courteous of the authors to mention the pre-existing literature referring to redox regulation of protein functions as potential drug target for apicomplexan parasites, which includes (but not limited to) work by the Becker and Muller groups in the malaria research field, and publications on the apicomplexns Toxoplasma gondii and Eimeria tenella.

The authors select an ER protein with Trx domain (no explanation provided for this selection, but should be) to start with. Genetic knockdown shows essentiality and microscopy is used to validate its ER location (though the section about trafficking signals is hard to follow for non-experts). IP experiments are used to discover putative partners and proposed clients, however an essential control is missing for this part (see below).

DVSF is used to demonstrate redox activity, however since this was not used before a validation is needed (see below). Co-IP post DVSF treatment is used to again capture putative redox interactors, focus on PDI8 and 11, and knock down demonstrated that PDI11 is essential. Reciprocal co-IP was used in attempt to validate J2/PDI8/BiP interactions however requires controls (see below). Finally, drugs discovered as human PDI inhibitors were shown to be toxic to Plasmodium and compound 16F16 reversed the observed DVSF cross-linking of J2, PDI8 and PDI11, leading to an extrapolation (see below) that redox partnerships are a good target for development of anti-malaria drugs.

**Part II – Major Issues: Key Experiments Required for Acceptance**

Reviewer #1: As it is, the conclusions go so far beyond the data that it is difficult to make suggestions of experiments. Likely, rewriting with significantly less speculative conclusions would make it easier to assess the paper.

Reviewer #2: More convincing data for the IP experiments (Supp Table I) is required including an example of how the "5-fold more abundant filter was applied.

Additional control are needed for the co-IP experiments (Fig 6).

Further evidence that 16F16 is on target is needed.

Reviewer #3: Major issues:

DVSF treatment is used to demonstrate redox activity and redox mediate interatcions, however since this method is used in the ER and in Plasmodium for the first time, it requires validation. How do we know that DVSF doesn’t simple link together any cysteine containing proteins in the very small ER of these organisms? A control whereby an ER protein with a cysteine known/expected no to be involved in disulfide exchange is necessary.

The identification of putative partners and clients via co-IP lacks rigor and a demonstration of functional relevance. The co-IP results described in line 179-198 find a broad range of ER proteins many of which may well be false positive. An appropriate control is to compare to a co-IP experiment with an unrelated tagged ER protein that is not expected to interact with the suggested partners or clients. PfMV would be prime candidate for such control according to Figure 3A.

Some clients found in said co-IP are highlighted in the result section and an extrapolation on the role of oxidative folding in their function is included in the discussion. This should be demonstrated. For example, the authors mentioned MSP1. A reciprocal co-IP to validate this proposed interaction should be included. And/or the authors can test if the J2 knock-down prevents function or biogenesis of MSP1. This is just an example. Some functional work is necessary for at least one proposed client.

The proposed interaction between J2/PDI8/BiP is not well supported. All the co-IPs shown in Figure 6 do not show the wash and there is no control to show that the IP material doesn’t contain unspecific interactors. To demonstrate co-IP that we should see that the protein of interest isn’t simply a contaminant of all fractions including wash, and that irrelevant proteins don’t contaminate the elution. In addition, even if evidence for co-IP upon DVSF treatment is provided, this still suffers from lack of control of whether the proteins that DVSF link together are true interaction partners in physiological conditions. Finally, the data obtained with DVSF is not consistent with the proposed interaction with BIP.

The proposed interference of drugs with redox interactions also lack rigor. Figure 7 B,C and D – it is important to show a protein stain of same gel run of treated parasites to conclude that it is a specific effect of blocking DVSF mediated cross-linking. Other measures of parasite biology should be included to conclude specific effect on ER oxidative folding as opposed to other defects. Even if those aspects are satisfied, the discussion of drug discovery potential should be less biased and highlight also caveats: disulfide exchange is a highly conserved mechanism posing a challenge to obtain specificity to one organism over another. What is known about the compounds selected, their toxicity in animal models, their potential therapeutic window etc etc

To enhance the functional aspect – assuming a client is validated as suggested above, interaction with said client should be disrupted by 16F16, which would provide further support to the proposed inhibition mechanism.

**Part III – Minor Issues: Editorial and Data Presentation Modifications**

Reviewer #1: (No Response)

Reviewer #2: (No Response)

Reviewer #3: Minor issues

- title focuses on “druggable” process, but only the last figure deals with drugs and needs confirmation (see above) this title is misleading

- the example of how Trx domains and PDIs work in general should go in the introduction

- pfMV used as ER marker – provide a reference for this

- the introduction should include more information about what was previously known: about redox regulation in apicomplexans, about J2, about the compounds used in Fig 7

- EC50 of aTc used as measure for inhibition of growth – is that the acceptable way? Do other Plasmodium research group that use this DOZI system measure growth in this way? If so, provide reference.

- hits from co-IP are prioritised based on presence of SP or 1TMD. This doesn't provide any certainty of ER localization or transit other compartments (mitochondria, peroxisome, plasma membrane) can be targeted this way without passing through the ER

- provide explanation to why PfPD11 was not identified in the first coIP but was trapped with DVSF?

– In Figure 3C the first protein band of lane 2 is not clear.

- CXXS is referred to as active site throughout the manuscript, however it is not a Trx active site.

- since the downregulation in FigS4 isn’t complete it is not supported that PDI11 is not essential

- figure 7B,C,D – it should be marked on the gels what bands were measured for making the graphs, all the gels included for statistics should be provided as supplementary.

- Supplementary figure 10 and Figure 7. It would be beneficial to include the stated IC50 values of the drugs against human cells for comparison, even though this is a proof of principle experiment

PLOS authors have the option to publish the peer review history of their article (what does this mean?). If published, this will include your full peer review and any attached files.

Reviewer #1: No

Reviewer #2: No

Reviewer #3: No
---

## [Decision Letter · Decision Letter 1]

8 Dec 2020

Dear Dr Muralidharan,

Thank you very much for submitting your manuscript "An essential redox network in the endoplasmic reticulum of malaria parasites is sensitive to small molecule inhibition" for consideration at PLOS Pathogens. As with all papers reviewed by the journal, your manuscript was reviewed by members of the editorial board and by several independent reviewers. The reviewers appreciated the attention to an important topic. Based on the reviews, we are likely to accept this manuscript for publication, providing that you modify the manuscript according to the review recommendations.

As you will see, two reviewers have raised minor points that require attention prior to the manuscript being considered further. Reviewer #1 requests that further clarification is provided regarding the criteria used to include proteins in the list of candidate partners identified in Figure 3. Could you please provide that clarification. Reviewer #3 requests that the title of the manuscript should be again modified in order to better describe its focus on characterisation of the PfJ2 redox network, rather than its potential as a drug target (which is indeed a relatively minor part of the work in terms of the number of figures devoted to it). While we believe this is a relatively less important issue than the first, we would still be grateful if you could give the point some consideration.

Sincerely,

Michael J Blackman

Associate Editor

PLOS Pathogens

Margaret Phillips

Section Editor

PLOS Pathogens

Kasturi Haldar

Editor-in-Chief

PLOS Pathogens

orcid.org/0000-0001-5065-158X

Michael Malim

Editor-in-Chief

PLOS Pathogens

orcid.org/0000-0002-7699-2064

As you will see, two reviewers have raised minor points that require attention prior to the manuscript being considered further. Reviewer #1 requests that further clarification is provided regarding the criteria used to include proteins in the list of candidate partners identified in Figure 3. Could you please provide that clarification. Reviewer #3 requests that the title of the manuscript should be again modified in order to better describe its focus on characterisation of the PfJ2 redox network, rather than its potential as a drug target (which is indeed a relatively minor part of the work in terms of the number of figures devoted to it). While we believe this is a relatively less important issue than the first, we would still be grateful if you could give the point some consideration.

Reviewer Comments (if any, and for reference):

Reviewer's Responses to Questions

**Part I - Summary**

Reviewer #1: Overall the paper is interesting and contributes significantly to our understanding of Trx proteins in the parasite. Whilst there are inevitably some aspects that might be followed up this is probably beyond the scope of the paper.

Figure 3 (list of proteins identified in pulldown) - the description in the main text remains problematic.

The overall significance of the pulldown in figure 3 remains unclear simply because the criteria used to select the proteins containing a signal sequence or TM domain strongly biases the list. It is unclear whether this data could really be interpreted if this criteria was not used i.e. without excluding the non-ss and non-TM proteins how many proteins would be enriched over the control.

Filtering the data in this way does not provide a high confidence list of proteins. As it is, it cannot really be interpreted by the reader (the reader cannot really tell if 5 ER proteins were enriched in the pulldown but also 100 cytoplasmic proteins). This need to be clarified.

Reviewer #2: The authors have addressed my queries and suggestions.

Reviewer #3: The revised manuscript now provides the critical controls for the main strategy used. Likewise the discussion in the context of drug discovery is revised. This satisfies my concerns.

**Part II – Major Issues: Key Experiments Required for Acceptance**

Reviewer #1: As above

Reviewer #2: (No Response)

Reviewer #3: The title still needs changing. The paper primarily describes cell biological characterisation, rather than a drug screen.

**Part III – Minor Issues: Editorial and Data Presentation Modifications**

Reviewer #1: As above

Reviewer #2: (No Response)

Reviewer #3: (No Response)

PLOS authors have the option to publish the peer review history of their article (what does this mean?). If published, this will include your full peer review and any attached files.

Reviewer #1: No

Reviewer #2: **Yes: **Leann Tilley

Reviewer #3: No
---

## [Editor Report · Decision Letter 2]

7 Jan 2021

Dear Vasant,

We are pleased to inform you that your manuscript 'A redox-active crosslinker reveals an essential and inhibitable oxidative folding network in the endoplasmic reticulum of malaria parasites' has been provisionally accepted for publication in PLOS Pathogens.

Best regards,

Michael J Blackman

Associate Editor

PLOS Pathogens

Margaret Phillips

Section Editor

PLOS Pathogens

Kasturi Haldar

Editor-in-Chief

PLOS Pathogens

orcid.org/0000-0001-5065-158X

Michael Malim

Editor-in-Chief

PLOS Pathogens

orcid.org/0000-0002-7699-2064
---

## [Editor Report · Acceptance letter]

29 Jan 2021

Dear Dr Muralidharan,

We are delighted to inform you that your manuscript, "A redox-active crosslinker reveals an essential and inhibitable oxidative folding network in the endoplasmic reticulum of malaria parasites," has been formally accepted for publication in PLOS Pathogens.

Best regards,

Kasturi Haldar

Editor-in-Chief

PLOS Pathogens

orcid.org/0000-0001-5065-158X

Michael Malim

Editor-in-Chief

PLOS Pathogens

orcid.org/0000-0002-7699-2064